# Self-protection soft fluidic robots with rapid large-area self-healing capabilities

Wei Tang [1,2,3], Yiding Zhong[1,2], Huxiu Xu[1,2], Kecheng Qin[1,2], Xinyu Guo[1,2], Yu Hu[1,2], Pingan Zhu[1,2], Yang Qu[1,2], Dong Yan[1,2], Zhaoyang Li[1,2], Zhongdong Jiao[1,2], Xujun Fan[1,2], Huayong Yang[1,2] & Jun Zou [1,2] ✉

Soft fluidic robots have attracted a lot of attention and have broad application prospects. However, poor fluidic power source and easy to damage have been hindering their development, while the lack of intelligent self-protection also brings inconvenience to their applications. Here, we design diversified self-protection soft fluidic robots that integrate soft electrohydrodynamic pumps, actuators, healing electrofluids, and E-skins. We develop high-performance soft electrohydrodynamic pumps, enabling high-speed actuation and large deformation of untethered soft fluidic robots. A healing electrofluid that can form a self-healed film with excellent stretchability and strong adhesion is synthesized, which can achieve rapid and large-areas-damage self-healing of soft materials. We propose multi-functional E-skins to endow robots intelligence, making robots realize a series of self-protection behaviors. Moreover, our robots allow their functionality to be enhanced by the combination of electrodes or actuators. This design strategy enables soft fluidic robots to achieve their high-speed actuation and intelligent self-protection, opening a door for soft robots with physical intelligence.

Soft fluidic robots[1,2] are the most prevalent in soft robotics due to their diversified design, large deformation, simple fabrication, and low cost, which are widely used in wearable devices, medical equipment, and industrial fields[3,4]. Since Joseph L. McKibben invented the first rudimentary soft fluidic robot (termed as McKibben artificial muscle) in 1950s[5], a great deal of research has been conducted on the design, actuation, sensing, control, and applications of soft fluidic robots[6]. Despite significant progress, poor fluidic power source[7] and easy to damage[8] have hindered the development of soft fluidic robots, while the lack of intelligent self-protection also brings inconvenience to their applications.

Most existing fluidic robots employ external bulky rigid pumps/compressors as tethered fluidic power sources, which limits their motions, whereas miniaturized rigid pumps that can be embedded into robots would limit their design and performance[9]. Some innovative methods, such as gas-liquid phase change[10], hydrogen

peroxide decomposition[11], explosion[12], etc., have been reported and effectively remove the fluidic tether, but they also introduce new challenges like slow actuation speed, reduced output force, or poor controllability[9]. Recent advances in electrohydrodynamic (EHD) technology[13,14], which manipulates dielectric liquids through electric fields, are used to manufacture EHD pumps to power soft fluidic robots with advantages such as lightweight, easy to embed, and silence. Existing EHD pumps like electro-conjugate fluid pump[15], stretchable pump[16], and soft electronic pump[17] possess excellent portability[15–17], good controllability[15–17], and some are easy to be customized[17]; however, when embedded into fluidic robots, their actuation performances are poor, resulting in slow actuation speed[15–18], tiny output force[15–18], limited actuation stroke[18,19], or difficult to achieve diversified design[19]. Although EHD pumps are expected to solve the challenge of fluidic power sources, their poor actuation performances limit their use in soft fluidic robots.

[1]State Key Laboratory of Fluid Power and Mechatronic Systems, Zhejiang University, Hangzhou 310027, China. [2]School of Mechanical Engineering, Zhejiang University, Hangzhou 310027, China. [3]Institute of Process Equipment, College of Energy Engineering, Zhejiang University, Hangzhou 310027, China. ✉e-mail: junzou@zju.edu.cn

Soft fluidic robots are at risk of damage, and healing the robot after a damage remains challenge[8,20]. The current research on self-healing of soft materials focuses on the self-healing of robot elastomer (i.e., elastomer self-healing)[20–25], which requires a long self-healing time and only realizes small-area damages[20], that is, the size of damages is smaller than the thickness of layers or the damage fracture surface is in contact. In the study of soft electronic pumps[17], we put forth the idea of using liquid to realize self-healing of soft materials (i.e., liquid to heal elastomer) and developed a healing liquid that can move in an electric field. When the robot was damaged, causing a liquid leakage, the healing liquid could cure when exposed to air to form a self-healed film that healed the breakage[17]. However, due to its poor adherence and limited stretching rate (~20%), this self-healed film can only repair small-area damages. Meanwhile, the long time (>6 h) needed for healing liquid restricts its application in soft fluidic robots. In addition, it is worthwhile to conduct research on self-protection (robot actively takes protective measures to achieve rapid self-healing when damaged), such as self-sensing, autonomous judge of damage, and self-heating to achieve rapid self-healing[20], for soft fluidic robots. Some soft devices have been reported for self-sensing[26], damage detecting[25], and heating to accelerate self-healing[27], but a soft device integrating self-sensing, self-detecting, and self-heating for soft fluidic robots has yet to be achieved.

## Results

### Design strategy of self-protection soft fluidic robots

The humans are typical soft fluidic systems[28,29] (Fig. 1a); they use the hearts as fluidic power sources, judge injury by skin sensing, and then actively take protective measures to achieve self-healing of large wounds through blood clotting. Inspired by human, we design a class of self-protection soft fluidic robots with untethered actuation and self-protection behaviors (Fig. 1a), where the ability of a robot to detect damage and then actively take protective measures to achieve rapid self-healing is termed as self-protection behavior. The soft EHD pump provides the hydraulic power that drives the healing electrofluid flow. Robot's motion is based on the power provided by the pump and the deformation of actuators. When the robot suffers a large damage, the E-skin sends resistance-change signal to the microchip and the robot would aware that it is damaged. Then, the robot will take some active protection to rapid deal with the damage, such as self-heating by E-skin. At last, the healing electrofluid will fill the large damage and solidify to achieve self-healing, with methyltracetoxysilane and dibutyltindilaurate in electrofluid acting as self-healing factors.

The design process of self-protection soft fluidic robots can be seen in Methods–"The design process of self-protection soft fluidic robots" and Supplementary Figs. 1–4. The advantages of flexible designs and diverse motions of conventional soft fluidic robots are largely retained by this design strategy, enabling the creation of self-protection soft fluidic robots for a variety of motions. To illustrate the design strategy of self-protection soft fluidic robots, we describe a design that generates bending motion (Fig. 1a), a bending soft fluidic robot that integrates a microchip, a soft EHD pump, a bidirectional bending actuator, an E-skin, and a battery, and is filled with a healing electrofluid. Both a microchip and a battery are integrated into the robot to exhibit its untethered actuation, where the robot can be controlled by a smartphone or a computer through WiFi to provide various outputs (Supplementary Movie 1). It is worth mentioning that the healing electrofluids we proposed combine electrical actuation with good healing capabilities, and the self-healed films they formed possess strong adhesion properties and excellent stretchability, allowing for large-area self-healing (the size of damage is larger than the thickness of layers or the fracture surface of damage is not in contact) of soft materials (silicone rubber) (Fig. 1b). In addition, the soft fluidic robot can be given intelligence and achieve a series of self-protection behaviors like human, including self-sensing (self-

detecting), self-judgment, self-heating, and rapid self-healing (Fig. 1c), by designing a liquid-metal-based E-skin in conjunction with the healing electrofluids.

The soft EHD pump achieves the EHD flow[17] of the healing electrofluids by generating an electric field through the electrode pairs (Fig. 1d), which enables pumping process. We propose an electrode pair dubbed a conical-array-porous-plate electrode pair (Fig. 1e) to obtain high-performance outputs of the soft EHD pump. The EHD flow effect is enhanced by using the conical array as the positive electrode of the pump and a porous plate as the negative electrode because the conical array can produce a stronger electric field than the previously described cylindrical electrode[17]. Since the conical array is difficult to fabricate by conventional casting method, we propose using multi-material 3D printing to construct the soft EHD pump, with conductive TPU for the electrodes and non-conductive TPU for the slots and supports (Fig. 1f and Supplementary Fig. 5). The printed conical array and porous plate electrodes are soft and bendable (Fig. 1e and Supplementary Fig. 5). The assembly accuracy of the pump is improved thanks to the higher manufacturing accuracy of 3D printing compared to casting, which also makes it possible to integrate a larger array of electrode pairs in series/parallel, further enhancing the output performance of the pump. Our experimental results indicate that the soft EHD pump created by the conical-array-porous-plate electrode pairs and multi-material 3D printing method achieves a fast system actuation speed (when the pump is embedded into a robot, <0.25 s, Supplementary Fig. 6) that is more than 120 times faster than the stretchable pump[16], more than 40 times faster than the electro-conjugate fluid pump[18], and more than 4 times faster than the soft electronic pump[17] (see Table 1 for comparison[15–18]). Supplementary Fig. 7 shows the pressure, flow rate, response time, and lifetime test of a soft EHD pump with two electrode pairs. Pumping the healing electrofluid between two chambers through the soft EHD pump allows the bending soft fluidic robot's deformation mechanism to operate (Methods–"Actuation principle and wireless control of self-protection soft fluidic robots" and Supplementary Fig. 8). By combining the functions of soft EHD pumps, actuators, E-skins, healing electrofluids, and multi-material 3D printing, this design strategy enables the implementation of a variety of motions, including bending, twisting, and contracting, as well as the self-protection of soft fluidic robots.

### High-speed actuation

For the three fundamental motions of bending, twisting, and contracting that are frequently present in existing robots, we designed corresponding soft fluidic robots and performed actuation tests (Supplementary Fig. 9). The bending soft fluidic robot can achieve bidirectional bending motion (Fig. 2a and Supplementary Movie 1), and its angle increases with the applied voltage (Supplementary Fig. 10a). The finite element simulation analysis of the robot (Fig. 2b, Methods–"Finite element simulation analysis" and Supplementary Fig. 11) is in good agreement with the experimental tests. The robot's bending angle in a 2-Hz voltage is shown in Fig. 2c, with a maximum bending angle of ~85°. A good consistency in the angle curve of the bending robot indicates that the robot responds well to electrical signals. To visually measure the angle of the twisting soft fluidic robot, two battens were attached to both sides of the robot (Fig. 2d and Supplementary Movie 2), where Fig. 2e and Supplementary Fig. 12 show its simulation analysis. As shown in Supplementary Fig. 10b, the robot's twisting angle increases with the applied voltage. Figure 2f shows twisting angle of the robot in a 2-Hz voltage, and the maximum twisting angle is ~20°. The testing and simulation of the contracting soft fluidic robot is shown in Fig. 2g, h, Supplementary Fig. 10c, Supplementary Fig. 13, and Supplementary Movie 3. The output stroke of the robot is depicted in Fig. 2i with a voltage of 1 Hz and a load of 20 g, with a maximum stroke of ~14 mm, which is a considerable actuation stroke in the field of soft robots. Supplementary Fig. 10d shows load

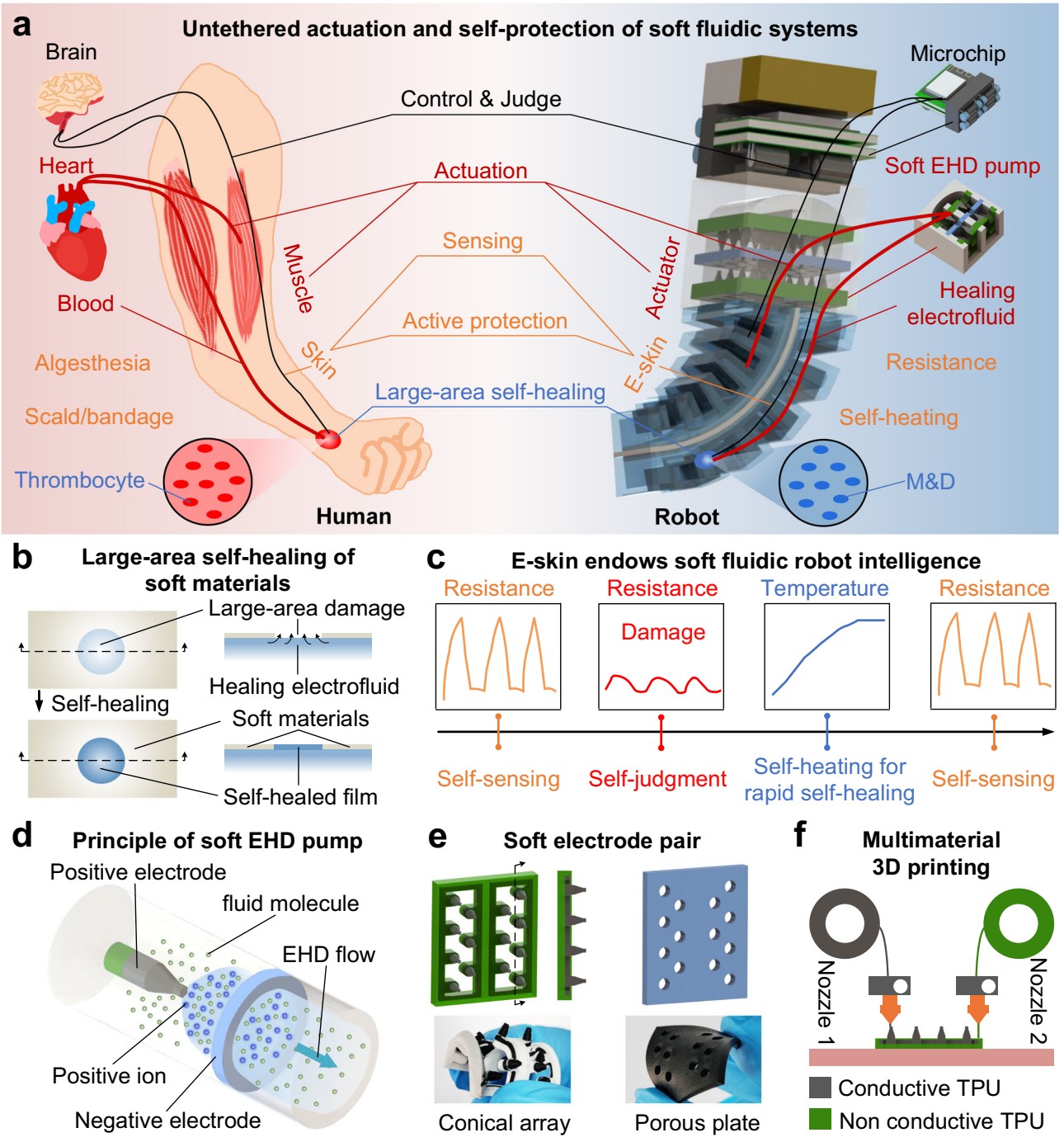

**Fig. 1 | Design strategy for self-protection soft fluidic robots. a** Untethered actuation and self-protection of soft fluidic systems. Human: control & judge−brain; actuation−heart, blood, muscle; sensing−skin, algesthesia; active protection−scald/bandage; large-area self-healing−thrombocyte. Robot: control & judge−on-board microchip; actuation−soft EHD pump, healing electrofluid, actuator; sensing−E-skin, resistance-change signal; active protection−self-heating by E-skin; large-area self-healing−methyltracetoxysilane and dibutyltindilaurate (M&D).

**b** Mechanism of large-area self-healing. The healing electrofluid fills the large-area damage of the soft material and then cures to firmly bond the soft material, thus achieving self-healing. **c** The E-skin combined with the healing electrofluid endow the fluidic robot intelligence, including a series of autonomous behaviors such as self-sensing, self-detecting, and self-heating for rapid self-healing. **d** Principle of the soft EHD pump. **e** Conical-array-porous-plate electrode pair. The electrodes are soft and bendable. **f** Multi-material 3D printing of the soft EHD pump.

tests of the robot, and the actuation stroke decreases with increasing load. The output force of the robot is -1 N. The actuation modes of our robots are not limited to these three modes; in fact, by carefully designing the structures of soft EHD pumps, actuators, and liquid reservoirs, it is easy to achieve various actuation modes including spiraling, radial expansion, and spatial bending. The matching of soft EHD pumps, actuators, and liquid reservoirs is the key to robot design, and different motions can be generated by changing the

structure of the actuator and following the design process in Supplementary Fig. 1.

A series of frequency tests were performed on the contracting robot to study the dynamic performance of the soft fluidic robot. To provide a restoring force required to return the robot, an elastic band was secured to the end of the robot (Fig. 2j). The dynamic performance of the robot was tested using voltage signals with an amplitude of 14 kV and frequencies ranging from 0.5 Hz to 1 kHz (Supplementary

**Table 1 | Comparison of different EHD pumps when the pumps are embedded into robots or actuators**

| | Actuation response | Stretchability | Force | Actuation stroke | Power consumption (actuation voltage) |
|---|---|---|---|---|---|
| Stretchable pump[16] | 30 s | Yes | \ | ~40° | ~0.1 W (8 kV) |
| Electro-conjugate fluid pump[18] | 10 s | No | ~0.18 N | ~1.5 mm | ~0.15 W (6 kV) |
| Electro-conjugate fluid pump[15] | 1.26 s | No | 0.0066 N | ~30° | \ (4.5 kV) |
| Soft electronic pump[17] | 1 s | Yes | \ | ~8 mm | ~3.6 W (16 kV) |
| Soft EHD pump (This study) | <0.25 s | Yes | >0.8 N | ~14 mm, ~85° | ~6 W (14 kV) |

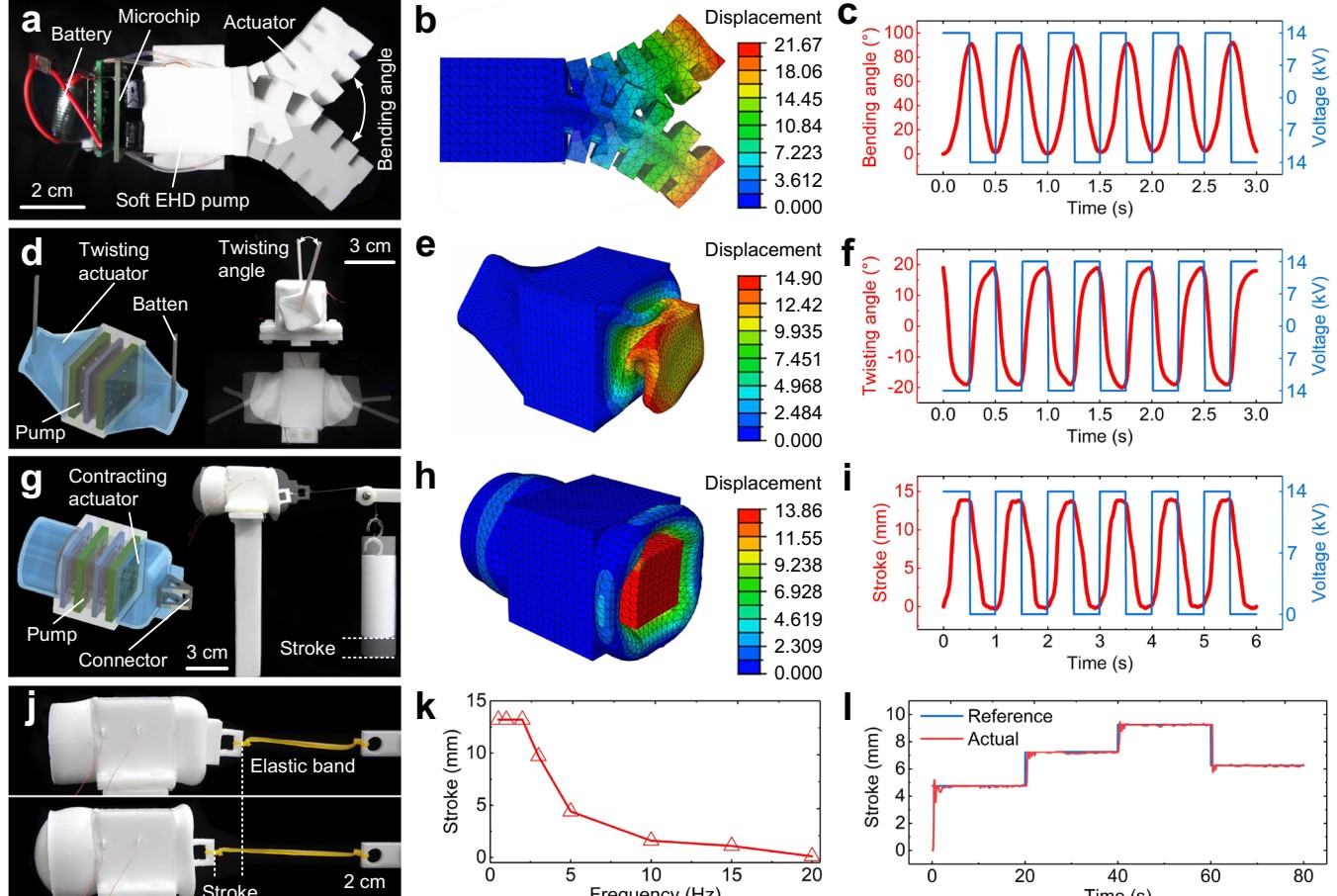

**Fig. 2 | High-speed actuation. a** Bending motion of the fluidic robot. The robot's electrodes are shown in Supplementary Fig. 5b, c. **b** Finite element simulation analysis of the bending fluidic robot. The color bar represents the displacement of the deformation, where the unit is mm. **c** Bending angle curve for the bending fluidic robot. The amplitude of the voltage is 14 kV, and the frequency is 2 Hz. The dielectric breakdown voltages of the EHD pump is -17 kV. **d** Twisting motion of the fluidic robot. The robot's electrodes are shown in Supplementary Fig. 5d, e. The angle between the two battens is zero while the robot is not in motion; it is recorded as a positive twisting angle when it generates counterclockwise twisting motion, and as a negative twisting angle when it generates clockwise twisting motion. **e** Finite element simulation analysis of the twisting fluidic robot. The color bar represents the displacement of the deformation, where the unit is mm. **f** Twisting angle curve for the twisting fluidic robot. The amplitude of the voltage is 14 kV, and the frequency is 2 Hz. **g** Contracting motion of the fluidic robot. The robot's electrodes are shown in Supplementary Fig. 5d, e. **h** Finite element simulation analysis of the contracting fluidic robot. The color bar represents the displacement of the deformation, where the unit is mm. **i** Actuation stroke curve for the contracting fluidic robot. The amplitude of the voltage is 14 kV, and the frequency is 1 Hz. **j** Frequency tests for the fluidic robot. The elastic band was secured to the end of the robot to provide a restoring force. **k** Frequency curve for the fluidic robot at a 14-kV applied voltage. **l** The stroke response curve for the fluidic robot under a PID closed-loop control.

Movie 4). The robot can generate a certain deformation within 0 Hz to 20 Hz (Fig. 2k), where the robot can maintain a large stroke within 2 Hz and a decreasing stroke with increasing frequency above 2 Hz. The robot produces a relatively small stroke at 50 Hz. As the frequency reaches 1 kHz, the robot's pump produces high-frequency vibration and the robot appears to have no apparent stroke. This type of fluidic robot greatly breaks the frequency bottleneck of traditional untethered soft fluidic robots, where the frequency of traditional untethered

soft fluidic robots with integrated miniature rigid pumps is usually ~1 Hz. Moreover, the frequency of the robot can be further increased by improving the structure of the actuator and using better electrofluids. A PID closed-loop control system (Methods—"PID closed-loop control system" and Supplementary Fig. 14) was built to verify the controllability of the fluidic robot. The stroke response curve (Fig. 2l) shows that the robot can be quickly stabilized at different deformation positions, indicating that the robot possess good controllability.

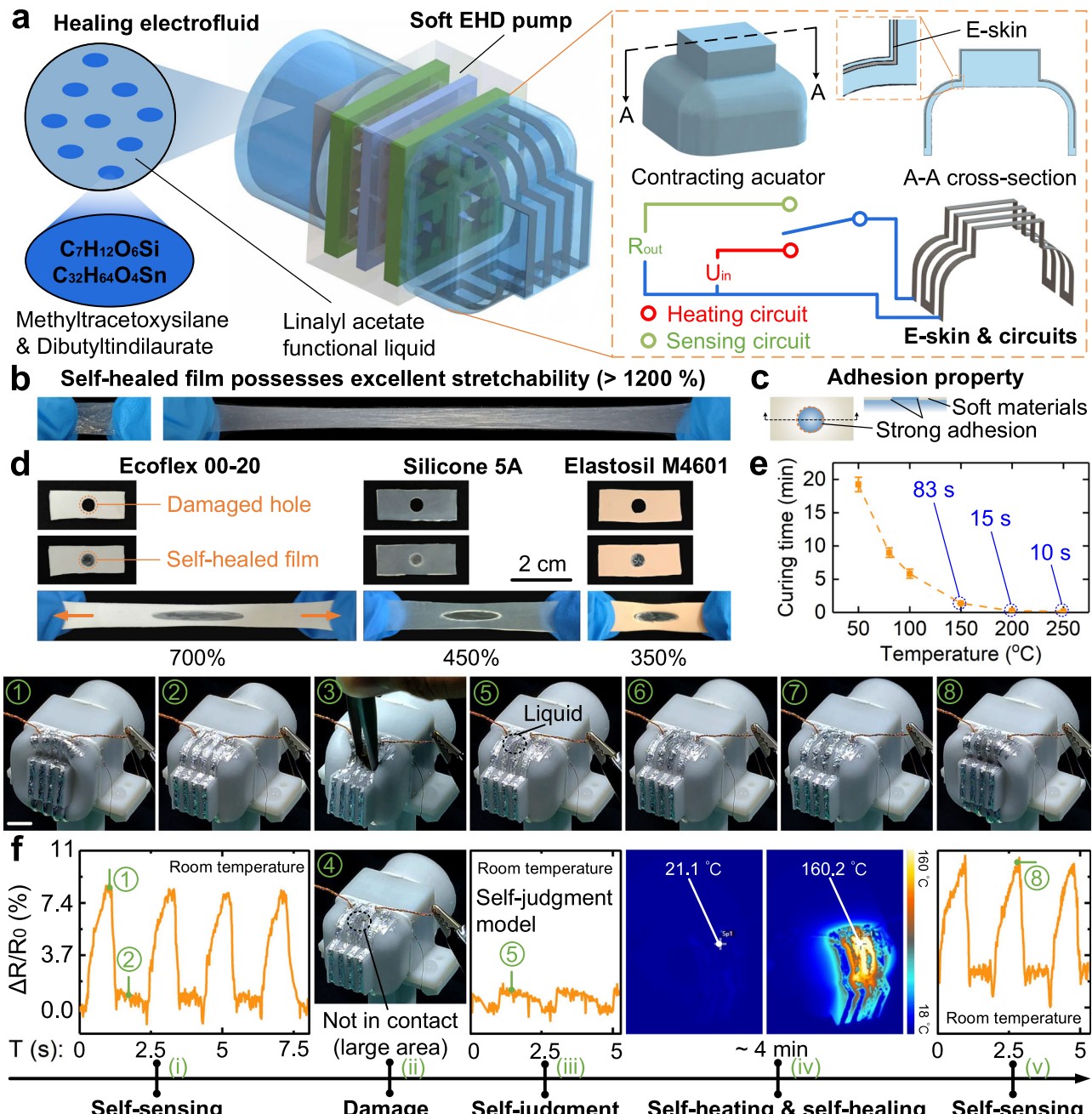

**Fig. 3 | Self-sensing, self-judgment, self-heating, and large-area self-healing of self-protection soft fluidic actuators. a** Architecture of a contracting soft fluidic robots. The robot's electrodes are shown in Supplementary Fig. 5d, e. The healing electrofluid is a mixture of linalyl acetate functional liquid, methyltracetoxysilane, and dibutyltindilaurate, which merges electrical actuation and good healing function. The multi-function E-skin integrates the functions of self-sensing, self-detecting, and self-heating. The circuits of the E-skin consist of a sensing circuit and a heating circuit. Although the E-skin has both heating and sensing functions, they are switched and cannot be utilized simultaneously, i.e., only as heating or sensing. **b** Excellent stretchability (>1200%) of self-healed films. **c** Strong adhesion of self-healed films with soft materials (silicone rubber). **d** Healing electrofluids are used to repair large-area damages in soft materials such as Ecoflex 00-20, Silicone 5A, and Elastosil M4601. **e** Curing time-temperature curve for the healing electrofluid. **f** The process of self-protection behaviors when the robot was damaged. The scale bar is 1 cm.

## Self-protection for rapid large-area self-healing

We utilized a contracting fluidic robot (Fig. 3a) as an illustration to demonstrate the robot's self-protection behaviors. Skin for self-sensing is essential for both humans and robots. An E-skin based on liquid metal is designed to endow the robot intelligence, where the E-skin can be embedded in the actuator or coated on the surface of the actuator (Methods−"Actuators and E-skins" and Supplementary Fig. 2). The E-skin deforms as the actuator is actuated, causing a change in the resistance value of the E-skin's output. The changing resistance value can be used to reflect the deformation state of the actuator, thus realizing the self-sensing of the robot (Methods−"Self-sensing of self-protection soft fluidic robots", Supplementary Fig. 15 and Supplementary Movie 5). The curves of the resistance value output from the E-skin are consistent and stable for the same actuation frequency (Supplementary Fig. 15), demonstrating that the E-skin can respond well to the deformation of the robot during the actuation process.

To solve the challenge of large-area self-healing for soft materials (silicone rubber), we explored and found that methyltracetoxysilane

($C_6H_{12}O_6Si$) and dibutyltindilaurate ($C_{32}H_{64}O_4Sn$) showed good mutual solubility with linalyl acetate functional liquid. Based on this finding, we report a kind of healing electrofluids (Fig. 3a and Methods −"Healing electrofluids"). When exposed to air, the healing electrofluid can cure into a self-healed film, which possesses excellent stretchability (>1200%, Fig. 3b) and strong adhesion properties (Fig. 3c). We tested the repair properties of the healing electrofluid on different silicone rubbers which are widely used in soft robots and found that the fluid could fill the large damage with no contact on the fracture surface, i.e., large-area damage, and achieve self-healing (Fig. 3d). When the three types of silicone rubbers were stretched to their corresponding maximum stretch rates without rupture (700%, 450%, and 350%), the self-healed film was still firmly bonded to the damage and maintained its functionality even when stretched to 700%. To further evaluate their mechanical properties, we tested stress-strain curves of pristine, 250 °C (15 s), damaged (hole 3–4 mm), and self-healed samples using Silicone 5 A as an example. The tensile stress-strain curves of the pristine and 250 °C (15 s) samples were similar, demonstrating that there was little variation between their mechanical properties (Supplementary Fig. 16a). Damaged samples were easily pulled off and had degraded mechanical properties, and after self-healing they could achieve similar mechanical properties to the pristine samples (Supplementary Fig. 16b). Compared with previously reported self-healing methods for soft materials, this method can achieve large-areas-damage self-healing of soft materials, which is difficult to achieve by solid self-healing methods of soft materials[20–25] and our previously reported liquid healing[17] (see Supplementary Table 1 for comparison[17,21,23,25,27,30,31]). Undoubtedly, our proposed large-areas-damage self-healing has some limitations. Healing electrofluid exposed to air can be cured to fill the damage, but if the damage is very large, it is difficult to heal the elastomer with a greater area of curing electrofluid. After many tests, we find that self-healing can be achieved at a distance of less than 5 cm of the fracture surface in damage. When the distance between the broken fracture surfaces is greater than 5 cm, the electrofluid curing tends to form holes that could lead to failure.

Inspired by the fact that scalding helps to accelerate the healing of human wounds, we found that temperature also had a significant impact on the curing time of the healing electrofluid (Fig. 3e), with a curing time of 83 s at 150 °C and 10 s at 250 °C. Higher temperatures can greatly shorten the healing time, leading us to consider in the direction of self-heating as active protection for robot. In addition to self-sensing, we found that the E-skin can also be utilized for heating, and its heating temperature can be stabilized at ~160 °C (Methods −"Self-heating for rapid large-area self-healing" and Supplementary Fig. 17). In addition, based on the self-sensing function, the E-skin could autonomously detect whether the robot was damaged or not (Methods−"Self-judgment of self-protection soft fluidic robots"). When the robot was damaged, the leaking liquid or incoming air bubbles had an effect on the deformation of the actuator, which could be reflected by the self-sensing of the E-skin. Based on a large amount of sensing data from both healthy and damaged robots, we developed a self-judgment model, allowing the robot to judge its own health state. When the robot judges that it is damaged, the E-skin automatically switches from self-sensing to self-heating function for self-protection so as to achieve rapid self-healing of the robot. It is worth noting that the liquid metal gives the E-skin itself good self-healing properties, which has been verified by relevant study[32].

We used a contracting soft fluidic robot to demonstrate its self-protection behaviors when damaged (Fig. 3f, Methods−"The self-protection process of a soft fluidic robot", and Supplementary Movie 6): (i) the robot worked normally and the E-skin output a normal sensing curve; (ii) the robot was damaged and the fracture surface of damage was not in contact, i.e., a large-area damage; (iii) the robot judged that it was at a damage state thanks to the self-judgment model; (iv) the robot automatically switched the E-skin from self-sensing to

self-heating state for self-protection, and the heating time was set to ~4 min in order to achieve sufficient self-healing (Methods−"Self-heating for rapid large-area self-healing" and Supplementary Fig. 17); (v) when the self-healing process was completed, the robot automatically switched the E-skin from self-heating to self-sensing state, and the E-skin output normally when the E-skin come down to room temperature; if the robot judged that it was still at a damage state, the step (iv) was repeated until the output was normal. The whole process shows the robot's intelligence and ability to implement self-protection behaviors like human, including self-sensing (self-detecting), autonomous self-judgment of damage, and autonomous self-heating for rapid large-area self-healing.

## Combination of electrodes/actuators for enhancing functionality

In addition to the intelligence for self-protection behaviors endowed by the E-skin and the healing electrofluid, the functionality of the robot can be enhanced by the combination of electrodes or actuators. The robot can reach a maximum stroke of 10 mm with a load of 50 g. After adding more electrode pairs, the robot can reach a load capacity of 330 g with a stroke of 9 mm (Fig. 4a and Supplementary Movie 7), demonstrating that the output force of the robot can be increased by integrating multiple electrode pairs, which is important for developing high-performance soft robots for large scale applications. Integrating two bending soft fluidic actuators with an on-board microchip and a battery creates an untethered soft gripper (Methods−"Untethered soft gripper" and Supplementary Fig. 18). The gripper could quickly catch a falling ping-pong ball which fell from a height at 65 cm/s (Fig. 4b and Supplementary Movie 8) using manually operated controls via a smartphone APP (Application), demonstrating that the gripper possesses high-speed actuation performance. The functionality of the mechanical sieve (Methods−"Mechanical sieve" and Supplementary Fig. 19) is made by multiple electrode pairs and actuators, where each of actuator can be programmed individually. The numbering of the actuators allows the use of a three-digit binary code to indicate the actuation state of the actuators, and the individual actuators can be programmed to be actuated sequentially or simultaneously. The actuators are sequentially controlled to realize the tilting rotation of the screening container, which throws the small beads out of the size-selective gate into the collection container, thus realizing the separation of the large beads from the small ones. The fact that our robots actuate mechanical sieves more quickly than traditional fluidic robots−requiring only 28 s (Fig. 4c and Supplementary Movie 9) as opposed to the 418 s previously described[33]−further demonstrates that our robots possess good dynamic performance and individual/sequential control performance. E-skin can be used for heat generation, while fluidic actuators can be used to generate tactile sensations. Combining the heating function of the E-skin and the actuation function of the fluidic actuator, we develop a tactile and thermal system (Fig. 4d). The system can give fingers tactile and thermal sensation (Fig. 4d), demonstrating the potential of soft fluidic robots for wearable devices.

## Discussion

The achievements described in this work represent a great advance in the development of soft fluidic robots. Soft EHD pumps and healing electrofluids we developed achieved fast untethered system actuation speed and large-areas-damage self-healing, addressing the challenges of poor fluidic power source and easy to damage. Our high-performance healing electrofluids, which can form a self-healed film with excellent stretchability (>1200%), strong adhesion, and rapid self-healing (10 s), offer a direction for large-areas self-healing of soft materials. Assisted by the multi-functional E-skin and microchip, our soft fluidic robots achieved a series of self-protection behaviors, including self-sensing (self-detecting), self-judgment, self-heating, and

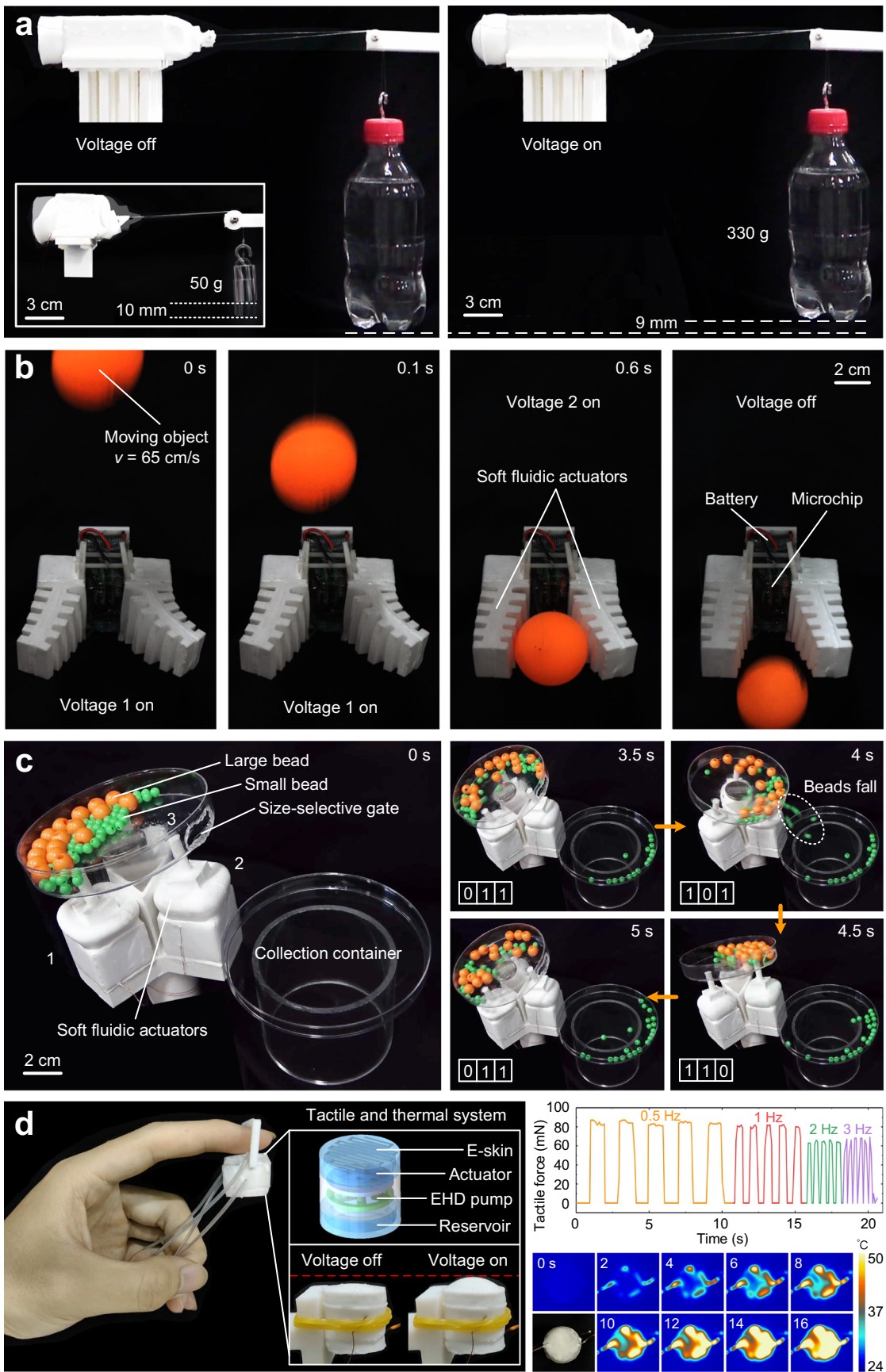

**Fig. 4 | Combination of electrodes/actuators for enhancing functionality. a** One robot pulls 50-g load and the other robot with more electrode pairs can pull 330-g load. **b** The untethered soft gripper can quickly grab a falling ping-pong ball which fell from a height at 65 cm/s using manually operated controls via a smartphone APP. **c** The mechanical sieve can separate the large beads from the small ones by the tilting rotation of the soft fluidic robot. **d** Tactile and thermal system.

rapid self-healing, which provides an idea for the development of intelligent soft robots and can be considered as one embodiment of physical intelligence in robot. The design of self-protection soft fluidic robots also allowed their functionality to be enhanced by the combination of electrodes or actuators. Compared with existing soft fluidic systems[10,15,16,19,21,25–27,34–40] (Supplementary Table 2), our design strategy of self-protection soft fluidic robot enables diversified design, high-speed actuation, self-sensing (self-detecting), self-judgment, and self-heating for rapid large-area self-healing, which is expected to bring prosperity to soft robots with physical intelligence[41,42].

## Methods

### The design process of self-protection soft fluidic robots

Self-protection soft fluidic robots consisted of four basic components: soft EHD pumps, actuators, E-skins, and healing electrofluids. Self-protection soft fluidic robots utilized EHD pumps as fluidic power source to actuate the actuators, resulting in producing desired motions. The detailed design process (Supplementary Fig. 1) was as follows: (i) design of actuators and E-skins; (ii) design of soft EHD pumps; (iii) finite element simulation analysis was used to optimize actuators and soft EHD pumps until they met the requirements; (iv) fabrication of self-protection soft fluidic robots. This design process is universal and can meet the design requirements of self-protection soft fluidic robots for various motions.

### Soft EHD pumps

Soft EHD pump consisted of three basic components: conical array electrodes, porous plate electrodes, and slots. The basic working principle of soft EHD pump was based on electrohydrodynamics, i.e., EHD flow[17] of the electrofluid generated by an inhomogeneous electric field. In order to improve the integration density and assembly accuracy of electrode pairs, we used multi-material 3D printing to fabricate the soft EHD pump. The soft EHD pump was mainly constructed from two types of soft materials, i.e., conductive and non-conductive materials, so we used a double nozzle 3D printer (E2, Raise3D) to implement this process. Conductive TPU (Eel, 90 A, NinjaTek) and non-conductive TPU (60 A, NinjaTek) were used as the printing materials for the pump. The individual electrode and slot of the pump were printed in parts and then assembled together to form the soft EHD pump. The 3D printing process was shown in Supplementary Fig. 5a, and the photos of the printed parts were shown in Supplementary Fig. 5b–g, where the resistances of the conical array electrode and porous plate electrode were ~4–10 kΩ and ~0.3–0.9 kΩ, respectively. These printed parts were soft and bendable (Supplementary Fig. 5). As long as the electrode pair was not shorted, soft EHD pumps could be deformed and still function, but if the deformation caused dielectric breakdown in the pumps, they would fail. Our experimental tests showed that the dynamic flow of the electrofluid did not have a significant effect on the electrode pairs. In Table 1, there are five different types of pumps: (i) stretchable pump[16]; (ii) electro-conjugate fluid pump[18]; (iii) electro-conjugate fluid pump[15]; (iv) soft electronic pump[17]; and (v) soft EHD pump (this study), where the electrodes of (i) are planar electrode structures and electrodes of (ii)-(v) are needle-hole electrode structures. Since the needle-hole electrode structure has a stronger ability to actuate the fluid flow compared to the planar electrode structure, (ii)-(v) have a larger flow rate than (i), and therefore the system responses of (ii)-(v) are faster than (i). (iii) integrated the actuator more tightly with the liquid reservoir, so (iii) has a faster system response than (ii). Since (iv) integrated more electrode pairs, the relative flow rate is increased, resulting in (iv) possessing a faster system response than (iii). The reason that (v) is better than (iv) is that, on the one hand, the conical electrode structure we propose can produce a stronger electric field strength than the cylindrical electrode structure, which increases the flow speed, and, on the other hand, we use 3D printing method to fabricate the electrodes so that more electrode pairs can be integrated, which increases the flow rate.

### Actuators and E-skins

Actuators were made of white Ecoflex 00-30 or white Ecoflex 00-20 (Smooth-on) and fabricated by mold-casting method. White Ecoflex 00-30/00-20 was prepared by mixing parts A and B in a 1:1 ratio and a small amount of white paste then degassing them in a vacuum dryer for ~3 min. Uncured white Ecoflex 00-30/00-20 was cast into the molds of the actuator and cured for ~6 h at room temperature, then removing the molds to form the actuator (Supplementary Fig. 2a). The E-skin was made of liquid metal, which was a gallium-indium-tin alloy with a melting point of 5 °C, and the mass fractions of gallium, indium, and tin were 62%, 25%, and 13%, respectively. The E-skin (Supplementary Fig. 2b) was formed by coating the actuator's outer surface with liquid metal. If the E-skin (Supplementary Fig. 2c) was to appear to be embedded in the deformation structure, it was then sealed by pouring uncured white Ecoflex 00-10 (Smooth-on) over the E-skin. E-skin without a seal can be refilled with liquid metal when there is a problem with the E-skin, but the exposed liquid metal is easily disturbed. E-skin with a seal is convenient to use, however, it is challenging to find E-skin issues when they occur.

### Healing electrofluids

Linalyl acetate functional liquid was an electrofluid with good electrical response. Methyltracetoxysilane ($C_6H_{12}O_6Si$) and dibutyltindilaurate ($C_{32}H_{64}O_4Sn$) (Silicone rubber adhesive, Sil-Poxy, Smooth-on) were high-performance solid bonding materials, which could firmly bond various silicone rubbers. Based on an accidental experiment, we found that methyltracetoxysilane and dibutyltindilaurate had good mutual solubility with linalyl acetate functional liquid. Based on this finding, we developed the healing electrofluid. The healing electrofluid was formed by dissolving methyltracetoxysilane and dibutyltindilaurate in dibutyl sebacate functional liquid by a magnetic stirrer, where their mass ratio was ~3:10. The viscosity, conductivity, and permittivity of the healing electrofluid were ~$1.116 \times 10^{-2}$ Pa·s (rotational viscometer), ~$1.99 \times 10^{-8}$ S/m (liquid conductivity meter), and ~4.2815 (vector network analyzer, coaxial reflection method, measurement frequency is 1 GHz), respectively. A temperature box with adjustable temperature was used to test the healing electrofluid's curing time at different temperature. Several healing electrofluids were positioned on the silicone sheets, and the silicone sheets were put in the temperature box. The self-healing region was periodically probed with a rod to evaluate whether the fluids had cured, and a timer was utilized to record the self-healing time at the same time. The time to failure of the healing electrofluid was about 10 days at room temperature in the laboratory. In addition, the electrofluid would swell silicone rubber. Silicone rubber tended to stabilize when swelling reached 5%, which had little effect on experimental testing. Swelling times to ~5% for different electrofluids are: linalyl acetate functional liquid, ~2 h; dibutyl sebacate functional liquid, ~20 days; and Fluorinert FC-40 functional liquid, >50 days. Improvement of electrofluid performance is an important direction for our follow-up research.

### Fabrication of self-protection soft fluidic robots

The detailed fabrication process of a self-protection soft fluidic robot (Supplementary Fig. 3) was described: (i) actuators and a soft EHD pump were bonded to form a soft structure with two chambers using white adhesive (704 white, Ausbond); (ii) the white adhesive was cured for ~3 h at room temperature and uncured white Ecoflex 00-20 was poured into the surface of the soft EHD pump to achieve better sealing. After that, an appropriate amount of healing electrofluid was filled into the chambers by using a syringe with a needle. Finally, the adhesive was applied on the punctured area of the needle to entirely seal the chambers, forming a self-protection soft fluidic robot.

## Actuation principle and wireless control of self-protection soft fluidic robots

The actuation principle of the self-protection soft fluidic robot was mainly based on the soft EHD pump and the actuators. The motions of the robot were achieved by pumping the liquid in two chambers through the soft EHD pump, so as to generate the deformation of the actuators. Supplementary Fig. 8 showed the actuation process of a bending soft fluidic robot. When voltage 1 was on and voltage 2 was off, the two electrode pairs on the soft EHD pump operated to pump the healing electrofluid from chamber 1 to chamber 2, resulting in an upward bending motion of the robot. Conversely, when voltage 2 was on and voltage 1 was off, the other two electrode pairs on the soft EHD pump worked to pump the healing electrofluid from chamber 2 to chamber 1, resulting in a downward bending motion of the robot. When both voltages were off, the healing electrofluid flowed back to its original location and the robot returned to its initial state. To wireless control of the self-protection soft fluidic robot, we integrated a WiFi module (ESP8266, ESP-12E) on the actuation circuit board and designed a smartphone APP based on this WiFi module using the Gizwits IoT cloud service platform (https://www.gizwits.com/). By connecting the smartphone to the robot through a local area wireless network, we could use this APP to realize the wireless control of the robot. The soft fluidic robots in Supplementary Movies 1–5 and 7 were controlled by a program that had been predetermined in the electronics or by a smartphone APP, where their system integration level can be seen in Supplementary Table 3. The actuation principle and wireless control for other types of self-protection soft fluidic robots was similar.

## Finite element simulation analysis

Abaqus was used for the finite element simulation analysis of the self-protection soft fluidic robots. The pressure generated by the soft EHD pump was applied to the actuators by equivalent means, i.e., Pressure in Abaqus was applied as a load on the inner surface of the actuators. Uniaxial tensile tests were conducted at an ASTM D638 (Type IV) universal test machine with a crosshead speed of 50 mm/min to obtain material properties of white Ecoflex 00-30 (Supplementary Fig. 4a) and white Ecoflex 00-20 (Supplementary Fig. 4b). A hyperelastic incompressible Yeoh material model[43], with strain energy $W = C_{10}(I_1-3) + C_{20}(I_1-3)^2 + C_{30}(I_1-3)^3$, was used to obtain the nonlinear material behavior of both white Ecoflex 00-30 and white Ecoflex 00-20. With the method of fitting (Supplementary Fig. 4), the material coefficients were $C_{10} = 0.0122$ MPa, $C_{20} = -0.0018$ MPa, $C_{30} = 0.0005$ MPa for white Ecoflex 00-30 and $C_{10} = 0.0093$ MPa, $C_{20} = -0.0017$ MPa, $C_{30} = 0.0004$ MPa for white Ecoflex 00-20. The finite element simulation analysis results of three kinds of the self-protection soft fluidic robots were shown in Supplementary Figs. 11–13.

## Experimental procedures of actuation tests

A custom LabVIEW program (Version 2021, 32-bit), a data acquisition board (DAQ, Model USB-6341, National Instruments), and a high-voltage amplifier (Model 20/20C, Trek) were used to generate the high voltage signals for the actuation tests of the self-protection soft fluidic robots, and an oscilloscope (InfiniiVision 2000 X-Series, Keysight) was used to display the voltage signals. The actuation test setups of the self-protection soft fluidic robots were shown in Supplementary Fig. 9. During the tests, the bottom surface of the actuator's soft EHD pump was fixed to the experiment table. The movies of the actuation tests were recorded using a camera, and the test results were measured using a custom movie processing program in MATLAB.

## PID closed-loop control system

A PID closed-loop control system was built to verify the controllability of the self-protection soft fluidic robot, as shown in Supplementary Fig. 14a. The system consisted of a computer, a MCU (Arduino UNO), an optical coupled isolation module (EL817), an analog to digital converter, a linear booster circuit, a miniature high-voltage power converter, a camera, and a contracting soft fluidic robot. This system was designed to verify the controllability of the robot, with the focus on controllability, so we used the camera as stroke feedback rather than the self-sensing function of the E-skin as stroke feedback. The PID control principle of the robot was shown in Supplementary Fig. 14b, where the robot was took as the controlled object, the stroke of the robot was took as the controlled variable, and the displacement captured by the camera was took as the feedback. The stroke response curve of the robot was shown in Fig. 2l. The PID controller could steadily control the robot at different deformation positions, demonstrating that the stroke of the robot was controllable.

## Self-sensing of self-protection soft fluidic robots

Liquid metals have been used in soft sensors and stretchable electronics due to their high electrical conductivity and tunable adhesion, and the changeable resistance in deformation region of liquid metals can be used as liquid metal-based strain sensor and E-skin[44]. The resistance of liquid metal-based E-skin can generate corresponding change along with its deformation[44,45]. When stretching or compressing the E-skin, the length of the liquid metal wires increases and the cross-sectional area decreases, and hence the resistance increases[44,45]. As the self-protection soft fluidic robot was actuated, the E-skin was deformed, causing the resistance value of the E-skin output to change. The changed resistance value could be used to respond to the deformation state of the actuator and thus for self-sensing of the robot. Under different actuation frequencies, the resistance output from the E-skin and the deformation state of the robot were shown in Supplementary Fig. 15. During the actuation process, the E-skin contracted when the fluid in the actuator flowed to the liquid reservoir, which corresponded to the robot state ①. When the fluid flowed to the actuator, the E-skin expanded outward, which corresponded to the robot state ⑥. ⑤, and ⑦ were the states of the actuator in the process of contraction and expansion, respectively. The amplitude change of the electrical resistance output from the E-skin depended on the deformation stroke of the robot. As the frequency increased, the deformation of the robot gradually became smaller, and therefore the amplitude of the electrical resistance output from the E-skin gradually became smaller. The curves of the resistance value output from the E-skin were consistent and stable for the same actuation frequency, indicating that the E-skin could respond well to the deformation of the robot during the actuation process.

## Self-judgment of self-protection soft fluidic robots

The robot's deformation amplitude was roughly constant at a constant actuation frequency, and the E-skin's corresponding resistance amplitude was also roughly constant. The resistance output from the E-skin was roughly constant at 0.5 Hz, 1 Hz, and 2 Hz, as illustrated Supplementary Fig. 15, and the resistance waveforms were steady over time. When the robot was damaged, the leaking liquid or incoming air bubbles had an effect on its deformation and could cause the EHD pump to short-circuit, resulting in the robot's failure, which could be reflected by the self-sensing of the E-skin. Based on a large amount of sensing data output from the E-skins of the robots where included healthy and damaged robots at different actuation frequencies, we found that the resistance amplitude of the E-skin is small for the damaged robot, which could barely generate deformation after it was damaged. In addition, we found that the robot started to fail when the resistance amplitude of its E-skin output was less than 60% of the normal resistance amplitude at a constant actuation frequency. As a result, we defined a damage threshold of 60% of the normal resistance amplitude at constant actuation frequency, indicating that the robot was considered damaged when its E-skin's resistance amplitude fell below that threshold, which was the self-judgment model. Based on the self-judgment model, the robot could judge its own health state.

## Self-heating for rapid large-area self-healing

The E-skin could be used for self-heating in addition to self-sensing. When a breakage was detected, the robot disconnected the self-sensing circuit of the E-skin and connected to the self-heating control circuit. Supplementary Fig. 17 showed the self-heating process of the E-skin, where the input power of the E-skin is ~4 W. The temperature gradually increased with the heating time and rose to ~160 °C in 3 min, after which the temperature of the E-skin stabilized at ~160 °C. From the temperature curve of the healing electrofluid (Fig. 3e), it was clear that the self-healing time was ~75 s at 160 °C. Hence, together with the previous 3-min heating process and the subsequent 1-min heating time at 160 °C, 4 min was selected as the heating time to satisfy the large-area self-healing process of the robot. The heating process facilitated the large-area self-healing process because the fluid would be lost if it was not cured for a long time, thus causing the self-healing process to fail. Therefore, if the self-healing process could not be achieved in a short time, large-area self-healing was difficult to achieve. About 20-ml of healing electrofluid was filled into the robot's chambers and about 1-ml fluid was used to heal the damage in Supplementary Movie 6. Although the E-skin has both heating and sensing functions, they are switched and cannot be utilized simultaneously, i.e., only as heating or sensing (Fig. 3a). The E-skin can self-heat up to 160 °C, at which temperature it is used to achieve rapid self-healing. And before switching to the sensing function when it's time to utilize it, it waits for the E-skin to come down to room temperature.

## The self-protection process of a soft fluidic robot

We used a contracting soft fluidic robot to demonstrate its self-protection behaviors when damaged (Fig. 3f and Supplementary Movie 6): (i) the robot worked normally and the E-skin output a normal sensing curve; the pump worked at a specific frequency and had no control, that is, the frequency given by the program in electronics. (ii) The robot was damaged and the fracture surface of damage was not in contact, i.e., a large-area damage; the robot was stopped manually to make damage with tweezers. (iii) The robot judged that it was at a damage state thanks to the self-judgment model; the E-skin was used as a sensor to output resistance and the electronics autonomously analyzed the amplitude of the resistance. (iv) The robot automatically switched the E-skin from self-sensing to self-heating state for self-protection, and the heating time was set to ~4 min in order to achieve sufficient self-healing (Supplementary Fig. 17); the E-skin's heating temperature and time were not controlled and were determined empirically, i.e., by using a 4-W power for 4 min and then stopping automatically; the heating time of 4 min was determined by the liquid cure time and experience. (v) When the self-healing process was completed, the robot automatically switched the E-skin from self-heating to self-sensing state, and the E-skin output normally when the E-skin come down to room temperature; the pump was manually turned on. Supplementary Movie 6 mainly demonstrates the autonomous self-protection process (self-detecting, self-judgment, self-heating, and self-healing) of the robot after damaged, we concentrate on the autonomy of this process rather than the robot's fully autonomous movement.

We tested the power consumption of all components in Supplementary Movie 6. The power consumption of electronics was <1 W, the power consumption of data transmission module was <1 W, the max power consumption of the soft EHD pump was ~6 W, the power consumption of sensor was <1 W, and the power consumption of heater was ~4 W. All components did not work at the same time and were only used when there was a demand for them, e.g., heating, which consumed a lot of power, was only used when there was a need for rapid self-healing. When only actuating the robot's deformations, a single battery (7.4 V, 350 mAh) could be used for 7–10 min. If the robot was damaged and needed self-protection, one battery was only enough for one cycle of operation (actuation, self-sensing (self-detecting), self-judgment, self-heating, and actuation, as shown in Supplementary Movie 6).

## Untethered soft gripper

The soft gripper consisted of two bending fluidic actuators, a microchip, a battery, and a frame, as shown in Supplementary Fig. 18. The bending fluidic actuator was a kind of electronically controlled actuators, so it was easy to integrate all components into an untethered system. The microchip was used to actuate and control the two bending actuators and the battery was used to power the microchip. The soft gripper was a completely untethered system and was directly controlled remotely by a smartphone via WiFi. The soft gripper was manually controlled via a smartphone APP. Supplementary Fig. 18b showed the actuation process of the soft gripper, where the gripper opened outward when voltage 1 was on, closed when voltage 2 was on, and reset when power was off.

## Mechanical sieve

The mechanical sieve (Fig. 4c) consisted of a fluidic robot, a screening container with a size-selective gate, large beads (diameter 10 mm), small beads (diameter 5 mm), and a collection container. The fluidic robot consisted of three actuators, several electrode pairs, and a reservoir cylinder. The size-selective gate was between the small bead and the large bead, thus ensuring that only the small beads could pass through the size-selective gate but not the large beads. The actuators were sequentially controlled to realize the tilting rotation of the screening container, which threw the small beads out of the size-selective gate into the collection container, thus realizing the separation of the large beads from the small ones. The robot of the mechanical sieve was shown in Supplementary Fig. 19. Three actuators were integrated in a fixed frame and shared the middle reservoir cylinder, and the three actuators and the bottom of the reservoir cylinder were connected. Three actuators were controlled independently and were programmed to sequence on and off. When voltage off, the actuators were in compression. When two actuators were activated but the third was not, the two activated actuators elongate while the third remained compressed, tilting the screening container to the side of the compressed actuator. Alternating two actuators activated and the third not activated, the screening container produced a continuous tilt rotation. To operate the mechanical sieve, we built a control circuit using an Arduino, three high-voltage power converters, and a battery, where the output intervals and frequency were controlled by a program. The mechanical sieve screening process was shown in Fig. 4c, noting the actuator in the activated state as 1, at which point the actuator elongated, and the unactivated state as 0, with a actuation interval of 0.5 s. Four intermediate consecutive moments were selected for analysis (Fig. 4c): at 3.5 s, the states of the three actuators were 0, 1, and 1, respectively, and the screening container was tilted toward actuator 1; at 4 s, actuator 1 was activated, actuator 2 was unactivated, actuator 3 remained in place, and the screening container was tilted toward actuator 2, which put small beads into the collection container through the size-selective gate; at 4.5 s, actuator 2 was activated, actuator 3 was unactivated, actuator 1 remained in place, and the screening container was tilted towards actuator 3 to mix the beads; at 5 s, it returned to the state of 3.5 s, mixing beads and starting the next cycle. Bead screening could be finished in ~28 s, and manual control was then used to complete this process.

## Data availability

The data that support the findings of this study are available in the main text and the Supplementary Information. The data are also available from the corresponding authors upon request.

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

## Acknowledgements

This work was supported by Zhejiang Provincial Natural Science Foundation of China under Grant no. LD22E050002 (J.Z.), International Cooperation Program of the Natural Science Foundation of China under Grant no.52261135542 (J.Z.), China National Postdoctoral Program for Innovative Talents under Grant no. BX20220267 (W.T.), and the China Postdoctoral Science Foundation under Grant no. 2023M733066 (W.T.).

## Author contributions

W.T. and J.Z. proposed and supervised the project. W.T. designed the research. W.T., Y.Z., H.X., K. Q., X.G., Y.H., P.Z., Y.Q., D.Y., Z.L., Z.J., and X.F. conducted the experimental work. W.T. developed the finite element simulation analysis and analyzed the data. W.T. wrote the manuscript with input from all authors. W.T., J.Z., and H.Y. revised the manuscript.

## Competing interests

The authors declare no competing interests.
