## [Peer Review File · Nature Communications]

Self-protection soft fluidic robots with rapid large-area self-healing capabilitiesREVIEWER COMMENTS

Reviewer #1 (Remarks to the Author):

The authors report on the realisation of a fluidic robot equipped with a mechanically soft electrohydrodynamic (EHD) pump and featuring self-sensing and self-healing performances. The robot is tested in several application scenarios including load lifting, catching of fast falling objects and object separation. The manuscript offers several appealing new technologies including the one to improve performance of EHD pumps relying on multimaterial 3D printing and realisation of electro-fluids with fast curing time. These technologies are relevant for future designs of soft robotic systems. Another important aspect of the manuscript that the robot can be turned autonomous and portable by using an on-board electronics and battery. The manuscript is prepared at high level with clear visual data and supporting video files.

I have the following comments, which should be clarified to make the manuscript clear from misinterpretations and allow reproducibility of the reported data:

1/ First paragraph on page 4 (and many similar occasions in the main text): although there is an analogy to the human body processes, I find the degree of details provided in the paragraph rather misleading and not needed for understanding of the manuscript. Hence, if nothing speaks against, I recommend to shorten this paragraph. It is sufficient to have figure 1(a) and a short remark in the text to explain the analogy.

2/ First paragraph on page 5: it is stated that the operation of the robot can be controlled wirelessly via smartphone and in the same sentence there is a reference to SI Movie 1. However, the movie does not show control of the robot by a smartphone. Do the authors use an app or how is this control done? More details would be needed.

3/ The reported results on EHD pumping are truly remarkable. The use of multimaterial 3D printing for fabrication is indeed an enabler of this work. I would invite the authors to extend their SI Table 1 (comparison table on EHD pumping systems) to benchmark not only the speed parameter but also stretchability, force, actuation stroke against the literature

data, e.g. REFs 15-19 and others. The comparison table should also include an estimate on the power consumption of the pump to support the statement of the authors on portability and autonomy of their robots.

4/ Discussion on actuation modes (bending, twisting...) could be extended and the key ideas related to the design of the robotic body to enable these distinct actuation modes should be provided (main text could include a brief summary of SI figures 8 and 9).

5/ In figure 2 to characterise the performance of their pump, the authors work with 14 kV. It would be of advantage to comment on why this voltage is chosen.

6/ Figure 2: it would be of advantage to provide more characterization data on how actuation stroke as well as bending and twisting angles depend on the applied voltage.

7/ Figure 2: it would be of advantage to quantify the experiment with lifting different weights shown in figure 2g. How does the actuation stroke depend on the weight and frequency?

8/ Figure 2: it would be of advantage to quantify the force of the actuator.

9/ Figure 2b,e,h: the colour bars have very small font, which makes them difficult to read? May be the authors could consider deleting some of the intermediate numbers on the colour bar but increase the font size? Furthermore, the colour bar should be explained in the figure caption. What is shown with colour?

10/ The discussion on the e-skin working principle could be extended and the authors could add a comment (and references) explaining why resistance change is observed with actuation.

11/ SI Figure 12: the amplitude change of the electrical resistance of e-skin is dependent on the actuation frequency. What is the reason for this dependence?

12/ Self-heating experiment: the temperature is driven to 160degC and kept at this level for some time. It would be of advantage to comment how the temperature is sensed? Do the authors use the e-skin to monitor temperature or they use thermocamera and the control is done manually? More in-depth discussion and precise wording would be needed.

13/ Judgment on the health state of the robot: the authors indicate that they developed an intelligent model to make the robot able to decide on its state. How is this realised and how is this decision making implemented in the untethered robot? Which parameter is analysed and how decision is taken? I assume that the authors analyse resistance of the e-skin. However, this resistance is changing with strain, temperature, actuation frequency. What is the approach towards the unambiguous judgement to discriminate between these factors and decide on the damage? The relevant section in methods "Self-judgment of self-protection soft fluidic robots" could be extended to demonstrate the training data sets and accuracy of the approach to predict the robot behaviour/state (I assume machine learning was used for predictive analysis).

14/ As a follow-up on the previous comment: Section on "Self-sensing of self-protection" in methods: "As the self-protection soft fluidic robot was actuated, the E-skin was deformed, causing the resistance value of the E-skin output to change." This criterium is not unique for the damage. As indicate in the comments above, the resistance changes upon actuation even during the normal robot operation but also with temperature and actuation frequency. Therefore, the use of the resistance readout to decide on the damage might be not appropriate. The authors should comment on the signal discrimination algorithms to enable autonomous robot operation.

15/ In the SI table 2 it would be important to indicate also temperature for all the mentioned studies at which healing was carried out.

16/ The authors report 10 sec self-healing time at 250degC. This temperature is rather high for elastomers. It would be important to report also on the stability of the mechanical properties of the robot after it is heated to 250degC.

This is a comment to figure 3e: it would be of advantage to demonstrate the mechanical

performance of the elastomers (as prepared without damage + when damaged and not repaired + when repaired after damage), say stress strain curves after elastomers are exposed to the indicated temperatures for relevant time.

17/ What is the current through the heater which is needed to reach 160degC or 250degC? How much power is needed to accomplish the healing process? Is this process compatible with autonomous and portable operation as announced even in the abstract (i.e., “autonomous rapid large-area self-healing”)?

General comment: The text is written with a certain degree of ambiguity: it can be (probably erroneously) concluded that every reported demo can be made autonomous and portable. It would be important to check the manuscript and revise it from any possible misinterpretation on autonomous operation and portability.

18/ For the demo(s) where it is appropriate, the authors need to comment on how long is the autonomy time of the robot considering the power consumption of the electronics (power consumption is not indicated), data transmission module (power consumption is not indicated), pump (power consumption is not indicated), sensors (power consumption is not indicated), heater (power consumption is not indicated)?

The battery needs to assure operation of the pump, sensor and heater. The discussion on autonomy should be carefully revised in the entire manuscript to avoid misunderstanding that all the tasks and functionalities reported in demonstrators are performed based on battery and on-board electronics.

19/ Figure 4b: I like the demo but find the discussion misleading as it is written now. Do the authors mean that the robot by itself detects the approach event of the ball and catches it? More clear description is needed to understand the demo.

20/ Section on Actuators and E-skins (line 435): melting point should be indicated with units “degrees Centigrade” or similar.

21/ Liquid metals are known to be stable at temperatures relevant for this work. Still, for completeness, the authors might want to add a comment on the behaviour of e-skin at elevated temperatures and provide relevant references.

22/ I find details provided on electro-fluid not sufficient to reproduce the reported results. The authors are invited to extend the discussion on the fabrication and especially on the characterization of the electro-fluids used in this work. What is the viscosity and electrical properties?

23/ The authors indicate that for self-healing “an appropriate amount of self-healing electro-fluid was filled into the chambers”. It is not clear how large are the chambers (volume) and how much liquid was used to heal the damage shown in the demo.

24/ The authors need to provide a more detailed explanation on the tasks taken care of by on-board electronics. The reported robot operation in different demos is rather complex. It would be of advantage to know if all these decisions are taken by the on-board electronics or the robot is externally (manually) controlled. Each experiment should be very carefully described (say in methods section) to avoid misunderstandings on what is done using on-board electronics and what is externally controlled.

For instance, for the demo with self-healing: the pump is working at specific frequency (how is it controlled?), the pump is stopped to make damage with tweezers (probably stopped manually but not by electronics), the sensor is measuring resistance (what is controlling and analysing the signal?), the heater is starting to heat and control the temperature (what is controlling this process?), the heater is off after certain time (how the decision is taken to stop the healing process?), the pump is again switched on (does the on-board electronic stops/starts the pump autonomously or this is done manually?).

Similar questions on the logic behind the operation of the robot I have for every demo. Naively (and probably not correct), I would assume that most of the decisions are done manually using external electronics. If on-board decision making is not very extensive, the authors might need to revise their statements on autonomous operation and portability.

Reviewer #2 (Remarks to the Author):

In this submission, Tang et al. present a series of system optimization compared to their previous work [17], and achieve a high level of device integration of soft EHD pumps, fluidic powered soft actuators, power electronics, and electronic skins for sensing signals. The authors demonstrate the self-healing capability and some other preliminary robotic manipulation demonstrations.

However, the authors tried too hard to impress the readers with not well-defined buzzwords (e.g., self-protection, self-judgment, and self-healing liquid), and superior performances (>50 Hz, rapid self-healing of large area damage etc.) but failed to explain what are the compromises. I personally find the story-telling/writing a bit misleading and difficult to appreciate which puts shadows on the actual engineering merits of this work.

1, I think when the authors say “self-healing electro-fluid”, they actually mean the electrofluid can heal soft elastomer materials. I would call the actuator or the robotic system self-healing, but not the electrofluid. How can a liquid heal itself? Same thing for self-healing blood, it is my first time to hear this expression.

The word “self-protection” is also a bit far-fetched. Protect the robot from what? The predators? Or humans or other robots? I suggest the authors think carefully to introduce new concepts and if they think it is necessary, please define it properly at its first appearance.

I also have some concerns when the authors describe the process of self-healing in the abstract. The blood forms clots to stop excess bleeding. The healing happens afterwards where new tissues are grown to restore the functions. And I would not describe here as “blood heals the wound” in human while human muscle is not a hydraulic actuator.

2, One feature that the authors highlighted is the dynamic performance of their soft robotic system. However, I cannot help noticing that the EHD pump takes quite a big volume fraction compared to the soft actuator. For a given hydraulic pump with a fixed pressure-flow diagram, if we reduce the soft actuator size, the required amount of liquid to inflate decreases, and the dynamic performance will improve. However, a very small soft actuator will also significantly limit the application scenarios for fluidic soft robots, especially considering the integration advantages for fluidic transmission.

In the paper, the authors did not exclude this important factor, but directly compared with other soft or stretchable pumps (in Extended data figure 1). Is this time constant an intrinsic feature of the pump? Or a feature of the actuator or combined? The authors need to consider the potential factors to provide a fair comparison.

I am not saying making small hydraulic actuators is not useful, but the expressions feel like it is an intrinsic feature of the system independent of other system parameters. If the dynamic performance is so important for the authors, I highly recommend the authors to show the dynamic responses of their soft pump and the typical pressure-flow rate curve so other researchers in the soft robotic community can understand the potential of this pumping system.

And if the authors insist on such modular soft robot systems with a big pump and small actuator, it is still ok. But then the authors are encouraged to explore their applications. For example, maybe demonstrate a mechanical display for blind people or a tactile interface between computer and human. I am sure there is potential and the authors can better promote their work to a bigger audience.

3, The self-healing capability of the soft actuator and the filling liquid is interesting but the investigation is a bit sloppy. Although the concept was reported in the previous publication, the authors made improvements so it can adapt to “large area” damage. (How large is large here?) However, the damage shown in Fig.3d on the real robotic actuator is quite different from punching a hole, am I right? There are so many ways the soft actuators can get damaged and the demonstration is too preliminary. After reading the experiments, I still don't fully understand or don't have enough experimental evidence to evaluate to what extent the self-healing will work and when it will not? Does self-healing impact on the dynamic performances of the soft robotic systems? The results are thin compared to the authors' strong claims. Personally, I cannot reach solid conclusions but only find the method promising. Maybe the authors can consider working on this.

4, The demonstrations in Fig.4 are nice but I would not consider it as a contribution in this work. They have been reported in other publications, and are quite far from real-world applications. But they certainly show the systems function well and I appreciated that.

5, In fact, there is quite some nice engineering work done. For example, the authors develop multi-material 3d printing to improve the electrode design. I am wondering how they are compared to the metal electrodes in [17]. Is there any problem with the long time performance of the conductive TPU? And to what extent is the pump can be deformed and still functional? Does soft material generate mismatched assembly positions between the cone electrodes and the pore positions, especially under the dynamic flow?

Overall I think the paper has some solid engineering but with an unfit story. It has many claims that are not well-defined and investigated in detail, and I am not sure some of the nice features are compatible. The authors are not well explained what are the compromises for those performances and what are the associated limitations. Maybe it is more acceptable in the material community but based on the current version, I would recommend it for a more specialized journal (or a material science journal).

Reviewer #3 (Remarks to the Author):

The manuscript reported a bio-inspired design strategy for self-protection soft fluidic robots and the soft robots integrate EHD pumps, actuators, self-healing electro-fluids, and E-skins. Poor fluidic power source, large area self-healing, and active self-protection of soft robots are persistent challenges in soft robotics. The important contributions made by the authors to address these challenges are exciting. The authors' development of rapid large area self-healing and the proposed concept of self-protection can greatly advance the development of soft robotics. Additionally, the authors significantly increased the EHD pump's response speed, which is far-reaching for the solution of actuating fluidic soft robotics. The manuscript is suitable for publication in Nature Communications after the following questions are addressed.

1. The hardness and resistivity of the conductive TPU used for 3D printing EHD pumps should to be characterized.
2. Please explain why the conical electrode the authors propose is better than the cylindrical electrode the authors previously reported.

3. The authors should add experiments and data on the system response of the EHD pumps.
4. The authors should do a comprehensive response speed comparison of more EHD pumps.
5. In Figure 2, why is there a negative angle in the twisting angle of the twisting actuator, and how is it defined?
6. Please clarify the advantages and disadvantages of the author's two methods for creating E-skin.
7. How is the self-healing liquid's curing time measured, and how is the curing curve created?
8. The authors seem to have different pumps for each soft robot, please clarify in detail the pumps used for different robots.
9. The sizes of the two types of beads used in the screening experiments need to be clarified. Also, please describe in detail how the mechanical sieve operates.

Detailed Responses to Reviewers

Reviewer #1: P1-P28

Reviewer #2: P29-P44

Reviewer #3: P45-P51

Reviewer #1 (Remarks to the Author):

In summary: The authors report on the realisation of a fluidic robot equipped with a mechanically soft electrohydrodynamic (EHD) pump and featuring self-sensing and self-healing performances. The robot is tested in several application scenarios including load lifting, catching of fast falling objects and object separation. The manuscript offers several appealing new technologies including the one to improve performance of EHD pumps relying on multimaterial 3D printing and realisation of electro-fluids with fast curing time. These technologies are relevant for future designs of soft robotic systems. Another important aspect of the manuscript that the robot can be turned autonomous and portable by using an on-board electronics and battery. The manuscript is prepared at high level with clear visual data and supporting video files.

I have the following comments, which should be clarified to make the manuscript clear from misinterpretations and allow reproducibility of the reported data:

Response:

Thank you very much for taking your valuable time to review our manuscript in your busy schedule. It is of great honor for us to get your professional and helpful comments to improve our manuscript. Thank you for your kind comments, we have added details to the paper to make the manuscript clear from misinterpretations and allow for reproducibility. According to your constructive comments, we have made corresponding modifications and added new necessary data/results. Point-by-point corrections/responses are listed below.

Comment 1

1/ First paragraph on page 4 (and many similar occasions in the main text): although there is an analogy to the human body processes, I find the degree of details provided in the paragraph rather misleading and not needed for understanding of the manuscript. Hence, if nothing speaks against, I recommend to shorten this paragraph. It is sufficient to have figure 1(a) and a short remark in the text to explain the analogy.

Answer 1

Thank you for your valuable comment. We shortened the description of human body processes and changed the description in the Main Text to a short remark (Marked in red on page 4). Also, we removed the description of the human body processes in the Abstract and the word “bio-inspired” in the manuscript.

In page 4, we have revised

Design strategy of self-protection soft fluidic robots

The humans are typical soft fluidic systems^{28,29} (Fig. 1a); they use the hearts as fluidic power sources, judge injury by skin sensing, and then actively take protective measures to achieve self-healing of large wounds through blood clotting.

Comment 2

2/ First paragraph on page 5: it is stated that the operation of the robot can be controlled wirelessly via smartphone and in the same sentence there is a reference to SI Movie 1. However, the movie does not show control of the robot by a smartphone. Do the authors use an app or how is this control done? More details would be needed.

Answer 2

Thank you for your helpful comment. On the actuation circuit board, we integrated a WiFi module (ESP8266, ESP-12E). Using the Gizwits IoT cloud service platform (<https://www.gizwits.com/>), we then designed a smartphone APP based on this WiFi module,

which can achieve the adjustment of actuation frequency. By connecting the smartphone to the robot through a local area wireless network, we use this APP to realize the wireless control of the robot. We have added the information in revised manuscript (Marked in red on page 21).

In page 21, we have added

To wireless control of the self-protection soft fluidic robot, we integrated a WiFi module (ESP8266, ESP-12E) on the actuation circuit board and designed a smartphone APP based on this WiFi module using the Gizwits IoT cloud service platform (<https://www.gizwits.com/>). By connecting the smartphone to the robot through a local area wireless network, we can use this APP to realize the wireless control of the robot.

Comment 3

3/ The reported results on EHD pumping are truly remarkable. The use of multimaterial 3D printing for fabrication is indeed an enabler of this work. I would invite the authors to extend their SI Table 1 (comparison table on EHD pumping systems) to benchmark not only the speed parameter but also stretchability, force, actuation stroke against the literature data, e.g. REFs 15-19 and others. The comparison table should also include an estimate on the power consumption of the pump to support the statement of the authors on portability and autonomy of their robots.

Answer 3

Thank you for your valuable comment. We have extended SI Table 1, added stretchability, force, actuation stroke, and power consumption, and added more EHD pumps in the literature. We have added the information in Supplementary information (Marked in red on Supplementary information page 21).

In Supplementary information page 21, we have revised

Supplementary Table 1 | Comparison of different EHD pumps when the pumps are embedded into robots or actuators.

	Actuation response	Stretchability	Force	Actuation stroke	Power consumption
Stretchable pump ¹⁶	30 s	Yes	\	~ 40°	~ 0.1 W
Electro-conjugate fluid pump ¹⁸	10 s	No	~ 0.18 N	~ 1.5 mm	~ 0.15 W
Electro-conjugate fluid pump ¹⁵	1.26 s	No	0.0066 N	~ 30°	\
Soft electronic pump ¹⁷	1 s	Yes	\	~ 8 mm	~ 3.6 W
Soft EHD pump (This study)	< 0.25 s	Yes	> 0.8 N	~ 14 mm ~ 85°	~ 6 W

Comment 4

4/ Discussion on actuation modes (bending, twisting...) could be extended and the key ideas related to the design of the robotic body to enable these distinct actuation modes should be provided (main text could include a brief summary of SI figures 8 and 9).

Answer 4

Thank you for your kind comment. We have added some discussions of actuation modes and robot design ideas in main text (Marked in red on page 8).

The actuation modes of our robots are not limited to these three modes; in fact, by carefully designing the structures of soft EHD pumps, actuators, and liquid reservoirs, it is easy to achieve various actuation modes including spiraling, radial expansion, and spatial bending. The matching of soft EHD pumps, actuators, and liquid reservoirs is the key to robot design, and different motions can be generated by changing the structure of the actuator and following the design process in Figure S1.

In page 8, we have added

The actuation modes of our robots are not limited to these three modes; in fact, by carefully designing the structures of soft EHD pumps, actuators, and liquid reservoirs, it is easy to achieve various actuation modes including spiraling, radial expansion, and spatial bending. The matching of soft EHD pumps, actuators, and liquid reservoirs is the key to robot design, and different motions can be generated by changing the structure of the actuator and following the design process in Supplementary Fig. 1.

Comment 5

5/ In figure 2 to characterise the performance of their pump, the authors work with 14 kV. It would be of advantage to comment on why this voltage is chosen.

Answer 5

Thank you for your kind comment. The higher the voltage, the better the actuation performance of the EHD pump, but the breakdown voltage of the EHD pump is ~ 17 kV. Hence, in order to be suitable for the use of the robot, we chose 14 kV as the actuation voltage.

In page 9, we have added

The dielectric breakdown voltages of the EHD pump is ~ 17 kV.

Comment 6

6/ Figure 2: it would be of advantage to provide more characterization data on how actuation stroke as well as bending and twisting angles depend on the applied voltage.

Answer 6

Thank you for your helpful comment. We have added tests of actuation stroke, bending angles (unilateral bending), and twisting angles with applied voltages (Marked in red on pages 7, 8, and Supplementary information page 11- Supplementary Figs. 10a-c).

In page 7, we have added

The bending soft fluidic robot can achieve bidirectional bending motion (Fig. 2a and

Supplementary Video 1), and its angle increases with the applied voltage (Supplementary Fig. 10a).

As shown in Supplementary Fig. 10b, the robot's twisting angle increases with the applied voltage.

In page 8, we have revised

The testing and simulation of the contracting soft fluidic robot is shown in Figs. 2g, h, Supplementary Fig. 10c, Supplementary Fig. 13, and Supplementary Video 3.

In Supplementary information page 11, we have added Supplementary Figs. 10a-c

Supplementary Fig. 10 | Static tests of the soft fluidic robots. a, Bending angle – voltage curve of the bending fluidic robot (unilateral bending). **b,** Twisting angle – voltage curve of

the twisting fluidic robot. **c**, Actuation stroke – voltage curve of the contracting fluidic robot. **d**, Actuation stroke – load curve of the contracting fluidic robot. The applied voltage is 14 kV.

Comment 7

7/ Figure 2: it would be of advantage to quantify the experiment with lifting different weights shown in figure 2g. How does the actuation stroke depend on the weight and frequency?

Answer 7

Thank you for your valuable comment. We have added the experiment with lifting different weights shown in figure 2g. The actuation stroke decreases with increasing load (Supplementary Fig. 10d). The actuation stroke can be maintain within 2 Hz and decreases with increasing frequency above 2 Hz (Fig. 2k). We have added the information in Main Text and Supplementary information (Marked in red on page 8 and Supplementary information page 11).

In page 8, we have added

Supplementary Fig. 10d shows load tests of the robot, and the actuation stroke decreases with increasing load.

where the robot can maintain a large stroke within 2 Hz and a decreasing stroke with increasing frequency above 2 Hz.

In Supplementary information page 11, we have added Supplementary Fig. 10d

Supplementary Fig. 10 | Static tests of the soft fluidic robots. a, Bending angle – voltage curve of the bending fluidic robot. **b,** Twisting angle – voltage curve of the twisting fluidic robot. **c,** Actuation stroke – voltage curve of the contracting fluidic robot. **d,** Actuation stroke – load curve of the contracting fluidic robot. The applied voltage is 14 kV.

Comment 8

8/ Figure 2: it would be of advantage to quantify the force of the actuator.

Answer 8

Thank you for your valuable comment. We tested the output force of the robot in figure 2 by a force meter and its output force at 14 kV was about 1 N. We have added the information in Main Text (Marked in red on page 8).

In page 8, we have added

The output force of the robot is ~ 1 N.

Comment 9

9/ Figure 2b,e,h: the colour bars have very small font, which makes them difficult to read? May be the authors could consider deleting some of the intermediate numbers on the colour bar but increase the font size? Furthermore, the colour bar should be explained in the figure caption. What is shown with colour?

Answer 9

Thank you for your valuable comment. We have deleted some of the intermediate numbers on the colour bar and increased the font size. The colour bar represents the displacement of the deformation (Marked in red on page 9).

In page 9, we have revised

Fig. 2 | High-speed actuation. **a**, Bending motion of the fluidic robot. The robot's electrodes are shown in Supplementary Figs. 5b and c. **b**, Finite element simulation analysis of the bending fluidic robot. The color bar represents the displacement of the deformation, where the unit is mm. **c**, Bending angle curve for the bending fluidic robot. The amplitude of the voltage is 14 kV, and the frequency is 2 Hz. The dielectric breakdown voltages of the EHD pump is ~ 17 kV. **d**, Twisting motion of the fluidic robot. The robot's electrodes are shown in Supplementary Figs. 5d and e. The angle between the two battens is zero while the robot is not in motion; it is recorded as a positive twisting angle when it generates counterclockwise twisting motion, and as a negative twisting angle when it generates clockwise twisting motion. **e**, Finite element simulation analysis of the twisting fluidic robot. The color bar represents the displacement of the deformation, where the unit is mm. **f**, Twisting angle curve for the twisting fluidic robot. The amplitude of the voltage is 14 kV, and the frequency is 2 Hz. **g**, Contracting motion of the fluidic robot. The robot's electrodes are shown in Supplementary Figs. 5d and e. **h**, Finite element simulation analysis of the contracting fluidic robot. The color bar represents the displacement of the deformation, where the unit is mm. **i**, Actuation stroke curve for the contracting fluidic robot. The amplitude of the voltage is 14 kV, and the frequency is 1 Hz. **j**, Frequency tests for the fluidic robot. The elastic band was secured to the end of the robot to provide a restoring force. **k**, Frequency curve for the fluidic actuator at a 14-kV applied voltage. **l**, The stroke response curve for the fluidic robot under a PID closed-loop control.

Comment 10

10/ The discussion on the e-skin working principle could be extended and the authors could add a comment (and references) explaining why resistance change is observed with actuation.

Answer 10

Thank you for your valuable comment. Based on some references, we explain the e-skin working principle and resistance change of the e-skin when actuation. We have added the information in Main Text (Marked in red on pages 23 and 32).

In page 23, we have added

Liquid metals have been used in soft sensors and stretchable electronics due to their high electrical conductivity and tunable adhesion, and the changeable resistance in deformation region of liquid metals can be used as liquid metal-based strain sensor and E-skin⁴⁴. The resistance of liquid metal-based E-skin can generate corresponding change along with its deformation^{44,45}. When stretching or compressing the E-skin, the length of the liquid metal wires increases and the cross-sectional area decreases, and hence the resistance increases^{44,45}.

In page 32, we have added

44. Wang, X., Guo, R. & Liu, J. Liquid Metal Based Soft Robotics: Materials, Designs, and Applications. *Adv. Mater. Technol.* **4**, 1800549 (2019).
45. Jiang, J., Zhang, S., Wang, B., Ding, H. & Wu, Z. Hydroprinted Liquid-Alloy-Based Morphing Electronics for Fast-Growing/Tender Plants: From Physiology Monitoring to Habit Manipulation. *Small* **16**, 2003833 (2020).

Comment 11

11/ SI Figure 12: the amplitude change of the electrical resistance of e-skin is dependent on the actuation frequency. What is the reason for this dependence?

Answer 11

Thank you for your valuable comment. The amplitude change of the electrical resistance output from the E-skin depends on the deformation stroke of the robot. As the frequency increases, the deformation of the robot gradually becomes smaller, and therefore the amplitude of the electrical resistance output from the E-skin gradually becomes smaller. We have added the information in Main Text (Marked in red on page 24).

In page 24, we have added

The amplitude change of the electrical resistance output from the E-skin depends on the deformation stroke of the robot. As the frequency increases, the deformation of the robot gradually becomes smaller, and therefore the amplitude of the electrical resistance output from the E-skin gradually becomes smaller.

Comment 12

12/ Self-heating experiment: the temperature is driven to 160degC and kept at this level for some time. It would be of advantage to comment how the temperature is sensed? Do the authors use the e-skin to monitor temperature or they use thermocamera and the control is done manually? More in-depth discussion and precise wording would be needed.

Answer 12

Thank you for your helpful comment. We didn't sense, monitor or control the temperature of the E-skin. Since this temperature is used to heat for rapid self-healing, it is only necessary to give a known temperature. Supplementary Fig. 13 showed the temperature of the e-skin that we measured using the infrared camera, and by using this temperature curve and the self-healing curve (Fig. 3e), we can get how long the heating time can realize the self-healing of the robot, i.e. ~ 4 min (The temperature gradually increased with the heating time and rose to ~ 160 °C in 3 minutes, after which the temperature of the E-skin stabilized at ~ 160 °C. From the temperature curve of the healing electro-fluid (Fig. 3e), it was clear that the self-healing time was ~ 75 s at 160 °C. Hence, together with the previous 3-minute heating process and the subsequent 1-minute heating time at 160 °C, 4 minutes was selected as the heating time to satisfy the large-area self-healing process of the robot.). The input power of the E-skin is ~ 4 W, and the maximum temperature it heated up to was ~ 160 °C. The E-skin's heating temperature and time were not controlled and were determined empirically, i.e. by using a 4-W power for 4 minutes and then stopping automatically; the heating time of 4 minutes was determined by the liquid cure time and experience. We have added the heating power information in Main Text and Supplementary information (Marked in red on pages 25, 26 and Supplementary information page 18).

In page 25, we have added

Supplementary Fig. 13 showed the self-heating process of the E-skin, where the input power of the E-skin is ~ 4 W. The temperature gradually increased with the heating time and rose to ~ 160 °C in 3 minutes, after which the temperature of the E-skin stabilized at ~ 160 °C. From

the temperature curve of the healing electro-fluid (Fig. 3e), it was clear that the self-healing time was ~ 75 s at 160 °C. Hence, together with the previous 3-minute heating process and the subsequent 1-minute heating time at 160 °C, 4 minutes was selected as the heating time to satisfy the large-area self-healing process of the robot.

In page 26, we have added

(iv) The robot automatically switched the E-skin from self-sensing to self-heating state for self-protection, and the heating time was set to ~ 4 min in order to achieve sufficient self-healing (Supplementary Fig. 17); the E-skin's heating temperature and time were not controlled and were determined empirically, i.e. by using a 4-W power for 4 minutes and then stopping automatically; the heating time of 4 minutes was determined by the liquid cure time and experience.

In Supplementary information page 18, we have added

Supplementary Fig. 17 | The heating process curve of the E-skin. The temperature gradually increases with the heating time and rises to ~ 160 °C in 3 minutes, after which the temperature of the E-skin stabilizes at ~ 160 °C. The input power of the E-skin is ~ 4 W.

Comment 13

13/ Judgment on the health state of the robot: the authors indicate that they developed an intelligent model to make the robot able to decide on its state. How is this realised and how is this decision making implemented in the untethered robot? Which parameter is analysed and

how decision is taken? I assume that the authors analyse resistance of the e-skin. However, this resistance is changing with strain, temperature, actuation frequency. What is the approach towards the unambiguous judgement to discriminate between these factors and decide on the damage? The relevant section in methods “Self-judgment of self-protection soft fluidic robots” could be extended to demonstrate the training data sets and accuracy of the approach to predict the robot behaviour/state (I assume machine learning was used for predictive analysis).

Answer 13

Thank you for your valuable comment. We analyse resistance of the E-skin. Although the e-skin has both heating and sensing functions, they are switched and cannot be utilized simultaneously, i.e., only as heating or sensing (Fig. 3a). The e-skin can self-heat up to 160 °C, at which temperature it is used to achieve rapid self-healing. And before switching to the sensing function when it's time to utilize it, it waits for the E-skin to come down to room temperature. Hence, temperature cannot disturb the resistance out from the E-skin.

The robot's deformation amplitude is roughly constant at a constant actuation frequency, and the E-skin's corresponding resistance amplitude is also roughly constant. The resistance output from the E-skin is roughly constant at 0.5 Hz, 1 Hz, and 2 Hz, as illustrated Supplementary Fig. 12, and the resistance waveforms are steady over time. When the robot was damaged, the leaking liquid or incoming air bubbles had an effect on its deformation and could cause the EHD pump to short-circuit, resulting in the robot's failure, which could be reflected by the self-sensing of the E-skin. Based on a large amount of sensing data output from the E-skins of the robots where included healthy and damaged robots at different actuation frequencies, we found that the resistance amplitude of the E-skin is small for the damaged robot, which could barely generate deformation after it was damaged. Additionally, we found that the robot started to fail when the resistance amplitude of its E-skin output was less than 60% of the normal resistance amplitude at a constant actuation frequency. As a result, we defined a damage threshold of 60% of the normal resistance amplitude at constant actuation frequency, indicating that the robot was considered damaged when its E-skin's resistance amplitude fell below that threshold, which was the self-judgment model. Based on the self-judgment model, the robot could judge its own health state.

We have extended the section in methods “Self-judgment of self-protection soft fluidic robots” and added the information in Main Text (Marked in red on pages 14, 24, and 25). In addition, to avoid misunderstandings, we label “room temperature” on Fig. 3f. To answer your question more visually, we make a schematic below.

In page 14, we have added

Although the E-skin has both heating and sensing functions, they are switched and cannot be utilized simultaneously, i.e., only as heating or sensing.

In pages 24-25, we have added

Self-judgment of self-protection soft fluidic robots

The robot's deformation amplitude is roughly constant at a constant actuation frequency, and the E-skin's corresponding resistance amplitude is also roughly constant. The resistance output from the E-skin is roughly constant at 0.5 Hz, 1 Hz, and 2 Hz, as illustrated Supplementary Fig. 12, and the resistance waveforms are steady over time. When the robot was damaged, the leaking liquid or incoming air bubbles had an effect on its deformation and could cause the EHD pump to short-circuit, resulting in the robot's failure, which could be reflected by the self-sensing of the E-skin. Based on a large amount of sensing data output from the E-skins of the robots where included healthy and damaged robots at different actuation frequencies, we found that the resistance amplitude of the E-skin is small for the damaged robot, which could barely

generate deformation after it was damaged. Additionally, we found that the robot started to fail when the resistance amplitude of its E-skin output was less than 60% of the normal resistance amplitude at a constant actuation frequency. As a result, we defined a damage threshold of 60% of the normal resistance amplitude at constant actuation frequency, indicating that the robot was considered damaged when its E-skin's resistance amplitude fell below that threshold, which was the self-judgment model. Based on the self-judgment model, the robot could judge its own health state.

In page 25, we have added

Although the E-skin has both heating and sensing functions, they are switched and cannot be utilized simultaneously, i.e., only as heating or sensing (Fig. 3a). The E-skin can self-heat up to 160 °C, at which temperature it is used to achieve rapid self-healing. And before switching to the sensing function when it's time to utilize it, it waits for the E-skin to come down to room temperature.

Comment 14

14/ As a follow-up on the previous comment: Section on “Self-sensing of self-protection” in methods: “As the self-protection soft fluidic robot was actuated, the E-skin was deformed, causing the resistance value of the E-skin output to change.” This criterium is not unique for the damage. As indicate in the comments above, the resistance changes upon actuation even during the normal robot operation but also with temperature and actuation frequency. Therefore, the use of the resistance readout to decide on the damage might be not appropriate. The authors should comment on the signal discrimination algorithms to enable autonomous robot operation.

Answer 14

Thank you for your helpful comment. The answer can be seen in Comment 13.

Comment 15

15/ In the SI table 2 it would be important to indicate also temperature for all the mentioned studies at which healing was carried out.

Answer 15

Thank you for your kind comment. We have added temperature in Supplementary Table 2 (Marked in red on Supplementary information page 22).

In Supplementary information page 22, we have added

Supplementary Table 2 | Comparison of different self-healing of soft materials.

	Type of damage and stretch rate	Time and temperature
Diels-Alder polymer ²¹	Only small-area, non-autonomous, ~ 100%	> 24 h, 80 °C
Photopolymerizable self-healing ²³	Only small-area, non-autonomous, ~ 1000%	> 24 h, 90 °C
Self-healing polyurethane urea elastomer ²⁵	Only small-area, autonomous, ~ 1100%	~ 6 h, 25 °C
Diels-Alder polymer ²⁷	Only small-area, autonomous, < 100%	> 24 h, 90 °C
Printed Diels-Alder polymer ³⁰	Only small-area, non-autonomous, ~ 100%	> 24 h, 120 °C
Healing agents ³¹	Only small-area, autonomous, non-stretchable	> 48 h, 25 °C
Self-healing liquid ¹⁷	Only small-area, autonomous, ~ 20%	> 6 h, 35 °C
Healing electro-fluid (This study)	Large-area, autonomous, > 1200%	10 s, 250 °C

Comment 16

16/ The authors report 10 sec self-healing time at 250degC. This temperature is rather high for

elastomers. It would be important to report also on the stability of the mechanical properties of the robot after it is heated to 250degC.

This is a comment to figure 3e: it would be of advantage to demonstrate the mechanical performance of the elastomers (as prepared without damage + when damaged and not repaired + when repaired after damage), say stress strain curves after elastomers are exposed to the indicated temperatures for relevant time.

Answer 16

According to your suggestion, we conducted the tensile stress-strain test of pristine, 250 °C (15 s), damaged, and self-healed samples. The tensile stress-strain curves of the pristine and 250 °C (15 s) samples were similar, demonstrating that there was little variation between their mechanical properties (Supplementary Fig. 16a). Damaged samples were easily pulled off and had degraded mechanical properties, and after self-healing they could achieve similar mechanical properties to the pristine samples (Supplementary Fig. 16b). We have added the information in Main Text and Supplementary information (Marked in red on page 11 and Supplementary information page 17).

In page 11, we have added

To further evaluate their mechanical properties, we tested stress-strain curves of pristine, 250 °C (15 s), damaged (hole 3-4 mm), and self-healed samples using Silicone 5A as an example. The tensile stress-strain curves of the pristine and 250 °C (15 s) samples were similar, demonstrating that there was little variation between their mechanical properties (Supplementary Fig. 16a). Damaged samples were easily pulled off and had degraded mechanical properties, and after self-healing they could achieve similar mechanical properties to the pristine samples (Supplementary Fig. 16b).

In Supplementary information page 17, we have added

Supplementary Fig. 16 | Tensile tests. To test the mechanical properties of samples (silicone 5A), uniaxial tensile tests were conducted at an ASTM D638 (Type IV) universal test machine with a crosshead speed of 50 mm/min. **a**, Tensile stress-strain curves of pristine and 250 °C (15 s) samples. **b**, Tensile stress-strain curves of pristine, damaged, and self-healed samples.

Comment 17

17/ What is the current through the heater which is needed to reach 160degC or 250degC? How much power is needed to accomplish the healing process? Is this process compatible with autonomous and portable operation as announced even in the abstract (i.e., “autonomous rapid large-area self-healing”)?

General comment: The text is written with a certain degree of ambiguity: it can be (probably erroneously) concluded that every reported demo can be made autonomous and portable. It would be important to check the manuscript and revise it from any possible misinterpretation on autonomous operation and portability.

Answer 17

Thank you for your valuable comment. We are given a heating power, the input power is ~ 4 W, the heating time is ~ 4 min, and a total power consumption of ~ 16 W is needed to complete the self-healing process. This is not a large energy consumption. This self-healing process is autonomous. We have checked the manuscript and revised it from any possible

misinterpretation on autonomous operation and portability.

In page 25, we have added

Supplementary Fig. 13 showed the self-heating process of the E-skin, where the input power of the E-skin is ~ 4 W.

Hence, together with the previous 3-minute heating process and the subsequent 1-minute heating time at 160 °C, 4 minutes was selected as the heating time to satisfy the large-area self-heating process of the robot.

Comment 18

18/ For the demo(s) where it is appropriate, the authors need to comment on how long is the autonomy time of the robot considering the power consumption of the electronics (power consumption is not indicated), data transmission module (power consumption is not indicated), pump (power consumption is not indicated), sensors (power consumption is not indicated), heater (power consumption is not indicated)?

The battery needs to assure operation of the pump, sensor and heater. The discussion on autonomy should be carefully revised in the entire manuscript to avoid misunderstanding that all the tasks and functionalities reported in demonstrators are performed based on battery and on-board electronics.

Answer 18

Thank you for your valuable comment. According to your suggestions, we tested the power consumption of all components. The power consumption of electronics is < 1 W, the power consumption of data transmission module is < 1 W, the max power consumption of the soft EHD pump is ~ 6 W, the power consumption of sensor is < 1 W, and the power consumption of heater is ~ 4 W.

All components do not work at the same time and are only used when there is a demand for them, e.g. heating, which consumes a lot of power, is only used when there is a need for rapid

self-healing. When only actuating the robot's deformations, a single battery (7.4 V, 350 mAh) could be used for 7-10 min. If the robot was damaged and needed self-protection, one battery was only enough for one cycle of operation (actuation, self-sensing, self-judgement, self-heating, and actuation, as shown in Supplementary Video 6).

We have added the information in Main Text (Marked in red on pages 26-27).

In pages 26-27, we have added

We tested the power consumption of all components in Supplementary Video 6. The power consumption of electronics was < 1 W, the power consumption of data transmission module was < 1 W, the max power consumption of the soft EHD pump was ~ 6 W, the power consumption of sensor was < 1 W, and the power consumption of heater was ~ 4 W. All components did not work at the same time and were only used when there was a demand for them, e.g. heating, which consumed a lot of power, was only used when there was a need for rapid self-healing. When only actuating the robot's deformations, a single battery (7.4 V, 350 mAh) could be used for 7-10 min. If the robot was damaged and needed self-protection, one battery was only enough for one cycle of operation (actuation, self-sensing (self-detecting), self-judgement, self-heating, and actuation, as shown in Supplementary Video 6).

According to your suggestions, we have carefully revised our discussion of autonomy in the entire manuscript.

Comment 19

19/ Figure 4b: I like the demo but find the discussion misleading as it is written now. Do the authors mean that the robot by itself detects the approach event of the ball and catches it? More clear description is needed to understand the demo.

Answer 19

Thank you for your valuable comment. The robot cannot detect the ball. The soft gripper was directly controlled remotely by a smartphone via WiFi. We manually controlled the soft gripper via a smartphone APP. To avoid misunderstanding, we have added clear description in Main

Text (Marked in red on pages 14, 17, and 27).

In page 14, we have revised

The gripper could quickly catch a falling ping-pong ball which fell from a height at 65 cm/s (Fig. 4b and Supplementary Video 8) using manually operated controls via a smartphone APP

In page 17, we have revised

Fig.4b, The untethered soft gripper can quickly grab a falling ping-pong ball which fell from a height at 65 cm/s using manually operated controls via a smartphone APP.

In page 27, we have added

The soft gripper was a completely untethered system and was directly controlled remotely by a smartphone via WiFi. The soft gripper is manually controlled via a smartphone APP.

Comment 20

20/ Section on Actuators and E-skins (line 435): melting point should be indicated with units “degrees Centigrade” or similar.

Answer 20

According to your suggestion, we use °C as the unit of melting point (Marked in red on page 19).

In page 19, we have revised

The E-skin was made of liquid metal, which was a gallium-indium-tin alloy with a melting point of 5 °C, and the mass fractions of gallium, indium and tin were 62%, 25% and 13%, respectively.

Comment 21

21/ Liquid metals are known to be stable at temperatures relevant for this work. Still, for completeness, the authors might want to add a comment on the behaviour of e-skin at elevated

temperatures and provide relevant references.

Answer 21

Thank you for your kind comment. I guess that our lack of detailed description caused you some misunderstanding. Liquid metals are used for sensing at room temperature, but not at high temperatures. Although the e-skin has both heating and sensing functions, they are switched and cannot be utilized simultaneously, i.e., only as heating or sensing (Fig. 3a). The e-skin can self-heat up to 160 °C, at which temperature it is used to achieve rapid self-healing. And before switching to the sensing function when it's time to utilize it, it waits for the E-skin to come down to room temperature. We have added the information in Main Text (Marked in red on pages 14 and 25).

In page 14, we have added

Although the E-skin has both heating and sensing functions, they are switched and cannot be utilized simultaneously, i.e., only as heating or sensing.

In page 25, we have added

Although the E-skin has both heating and sensing functions, they are switched and cannot be utilized simultaneously, i.e., only as heating or sensing (Fig. 3a). The E-skin can self-heat up to 160 °C, at which temperature it is used to achieve rapid self-healing. And before switching to the sensing function when it's time to utilize it, it waits for the E-skin to come down to room temperature.

Comment 22

22/ I find details provided on electro-fluid not sufficient to reproduce the reported results. The authors are invited to extend the discussion on the fabrication and especially on the characterization of the electro-fluids used in this work. What is the viscosity and electrical properties?

Answer 22

Thank you for your valuable comment. The viscosity, conductivity, and permittivity of the healing electro-fluid were $\sim 1.116 \times 10^{-2} \text{ Pa}\cdot\text{s}$ (rotational viscometer), $\sim 1.99 \times 10^{-8} \text{ S/m}$ (liquid conductivity meter), and ~ 4.2815 (vector network analyzer, coaxial reflection method, measurement frequency is 1 GHz), respectively. We have added the information in Main Text (Marked in red on page 20).

In page 20, we have added

The viscosity, conductivity, and permittivity of the healing electro-fluid were $\sim 1.116 \times 10^{-2} \text{ Pa}\cdot\text{s}$ (rotational viscometer), $\sim 1.99 \times 10^{-8} \text{ S/m}$ (liquid conductivity meter), and ~ 4.2815 (vector network analyzer, coaxial reflection method, measurement frequency is 1 GHz), respectively.

Comment 23

23/ The authors indicate that for self-healing “an appropriate amount of self-healing electro-fluid was filled into the chambers”. It is not clear how large are the chambers (volume) and how much liquid was used to heal the damage shown in the demo.

Answer 23

Thank you for your valuable comment. About 20-ml of healing electro-fluid was filled into the robot's chambers and about 1-ml fluid was used to heal the damage in Supplementary Video 6. We have added the information in Main Text (Marked in red on page 25).

In page 25, we have added

About 20-ml of healing electro-fluid was filled into the robot's chambers and about 1-ml fluid was used to heal the damage in Supplementary Video 6.

Comment 24

24/ The authors need to provide a more detailed explanation on the tasks taken care of by on-board electronics. The reported robot operation in different demos is rather complex. It would

be of advantage to know if all these decisions are taken by the on-board electronics or the robot is externally (manually) controlled. Each experiment should be very carefully described (say in methods section) to avoid misunderstandings on what is done using on-board electronics and what is externally controlled.

For instance, for the demo with self-healing: the pump is working at specific frequency (how is it controlled?), the pump is stopped to make damage with tweezers (probably stopped manually but not by electronics), the sensor is measuring resistance (what is controlling and analysing the signal?), the heater is starting to heat and control the temperature (what is controlling this process?), the heater is off after certain time (how the decision is taken to stop the healing process?), the pump is again switched on (does the on-board electronic stops/starts the pump autonomously or this is done manually?).

Similar questions on the logic behind the operation of the robot I have for every demo. Naively (and probably not correct), I would assume that most of the decisions are done manually using external electronics. If on-board decision making is not very extensive, the authors might need to revise their statements on autonomous operation and portability.

Answer 24

Thank you for your valuable comment. There is no information in Supplementary Videos 1-5 and 7 that requires a decision, where the frequency change is managed by a program that has been predetermined in the electronics or by a smartphone APP. We have added the information in Main Text (Marked in red on page 21).

In page 21, we have added

The soft fluidic robots in Supplementary Videos 1-5 and 7 are controlled by a program that has been predetermined in the electronics or by a smartphone APP.

Supplementary Video 6 details the process as follows:

The pump is working at a specific frequency and has no control, that is, the frequency given

by the program in electronics. The pump is stopped manually to make damage with tweezers. The E-skin is used as a sensor to output resistance and the electronics autonomously analyzes the amplitude of the resistance. Heating is switched autonomously, when the robot autonomously detects a damage, the E-skin is switched autonomously from sensing to heating. The E-skin's heating temperature and time are not controlled and are determined empirically, i.e. by using a 4-W power for 4 minutes and then stopping automatically. The heating time of 4 minutes is determined by the liquid cure time and experience. The pump is manually turned on. The pump is started and stopped manually in this process; all other actions, such as detecting, judgment, heating, and healing, are autonomous. This demo mainly demonstrates the autonomous self-protection process (self-detecting, self-judgment, self-heating, and self-healing) of the robot after damaged, we concentrate on the autonomy of this process rather than the robot's fully autonomous movement. We have added the information in Main Text (Marked in red on page 26 - Methods).

In page 26, we have added

The self-protection process of a soft fluidic robot

We used a contracting soft fluidic robot to demonstrate its self-protection behaviors when damaged (Fig. 3f and Supplementary Video 6): (i) the robot worked normally and the E-skin output a normal sensing curve; the pump worked at a specific frequency and had no control, that is, the frequency given by the program in electronics. (ii) The robot was damaged and the fracture surface of damage was not in contact, i.e., a large-area damage; the robot was stopped manually to make damage with tweezers. (iii) The robot judged that it was at a damage state thanks to the self-judgment model; the E-skin was used as a sensor to output resistance and the electronics autonomously analyzed the amplitude of the resistance. (iv) The robot automatically switched the E-skin from self-sensing to self-heating state for self-protection, and the heating time was set to ~ 4 min in order to achieve sufficient self-healing (Supplementary Fig. 17); the E-skin's heating temperature and time were not controlled and were determined empirically, i.e. by using a 4-W power for 4 minutes and then stopping automatically; the heating time of 4 minutes was determined by the liquid cure time and

experience. (v) When the self-healing process was completed, the robot automatically switched the E-skin from self-heating to self-sensing state, and the E-skin output normally when the E-skin come down to room temperature; the pump was manually turned on. Supplementary Video 6 mainly demonstrates the autonomous self-protection process (self-detecting, self-judgment, self-heating, and self-healing) of the robot after damaged, we concentrate on the autonomy of this process rather than the robot's fully autonomous movement.

Decision making for video 8 and video 9 is controlled externally. We describe the operation of Video 9 (mechanical sieve) in detail. We have added the information in Main Text (Marked in red on pages 14, 27, and 28).

In page 14, we have added

(Fig. 4b and Supplementary Video 8) using manually operated controls via a smartphone APP

In page 27, we have added

The soft gripper is manually controlled via a smartphone APP.

In page 28, we have added

To operate the mechanical sieve, we built a control circuit using an Arduino, three high-voltage power converters, and a battery, where the output intervals and frequency are controlled by a program. The mechanical sieve screening process is shown in Fig. 4c, noting the actuator in the activated state as 1, at which point the actuator elongates, and the unactivated state as 0, with a actuation interval of 0.5 s. Four intermediate consecutive moments were selected for analysis (Fig. 4c): at 3.5 s, the states of the three actuators were 0, 1, and 1, respectively, and the screening container was tilted toward actuator 1; at 4 s, actuator 1 was activated, actuator 2 was unactivated, actuator 3 remained in place, and the screening container was tilted toward actuator 2, which put small beads into the collection container through the size-selective gate; at 4.5 s, actuator 2 was activated, actuator 3 was unactivated, actuator 1 remained in place, and the screening container was tilted towards actuator 3 to mix the beads; at 5 s, it returned to the

state of 3.5 s, mixing beads and starting the next cycle. Bead screening can be finished in ~ 28 s, and manual control was then used to complete this process.

According to your suggestions, we have carefully revised our statements on autonomous operation and portability in the entire manuscript.

Reviewer #2 (Remarks to the Author):

In summary: In this submission, Tang et al. present a series of system optimization compared to their previous work [17], and achieve a high level of device integration of soft EHD pumps, fluidic powered soft actuators, power electronics, and electronic skins for sensing signals. The authors demonstrate the self-healing capability and some other preliminary robotic manipulation demonstrations.

However, the authors tried too hard to impress the readers with not well-defined buzzwords (e.g., self-protection, self-judgment, and self-healing liquid), and superior performances (>50 Hz, rapid self-healing of large area damage etc.) but failed to explain what are the compromises. I personally find the story-telling/writing a bit misleading and difficult to appreciate which puts shadows on the actual engineering merits of this work.

Response:

Thank you very much for taking your valuable time to review our manuscript in your busy schedule. We feel great thanks for your professional and valuable comments to improve our manuscript. According to your constructive comments, we have made corresponding modifications and added new experimental data/results. In order to avoid any misleading, we defined/revised some terms, revised the story-telling/writing of the manuscript, and shortened the description of human body processes. Point-by-point corrections/responses are listed below. Thanks again!

Comment 1

1, I think when the authors say “self-healing electro-fluid”, they actually mean the electrofluid can heal soft elastomer materials. I would call the actuator or the robotic system self-healing, but not the electrofluid. How can a liquid heal itself? Same thing for self-healing blood, it is my first time to hear this expression.

The word “self-protection” is also a bit far-fetched. Protect the robot from what? The predators? Or humans or other robots? I suggest the authors think carefully to introduce new concepts and if they think it is necessary, please define it properly at its first appearance.

I also have some concerns when the authors describe the process of self-healing in the abstract. The blood forms clots to stop excess bleeding. The healing happens afterwards where new tissues are grown to restore the functions. And I would not describe here as “blood heals the wound” in human while human muscle is not a hydraulic actuator.

Answer 1

Thank you for your constructive comment. We agree with you. The electro-fluid is used to heal soft materials. The actuator or the robotic system can be called as self-healing system. The liquid cannot heal itself.

We borrowed from some references (R. Rhodes, S. Basu, I. German, G.C. Stevens, Self-healing electrical insulation systems, 2017 INSUCON - 13th International Electrical Insulation Conference (INSUCON), Birmingham, UK, 18678950, 2017; S. Basu, I. German, R. Rhodes, G.C. Stevens, J. Thomas, Advanced materials for self-healing electrical insulation systems, Advanced Materials - TechConnect Briefs 2016, 153 – 156, 2016), they used fluids to heal cables and defined the fluids as self-healing fluids or self-healing cable oils, we used proposed electro-fluid to heal soft elastomer materials, there are some similarities, so we followed their terminology and defined proposed electro-fluid as self-healing electro-fluid.

To avoid ambiguity, we have modified the term “self-healing electro-fluid” to “healing electro-fluid”, which is defined as that healing electrofluid is used to heal soft materials. We have revised throughout the manuscript.

In fact, blood plays an important role in the self-healing of human skin, especially its healing factor – thrombocyte. Similarly, healing factor - methyltracetoxysilane and dibutyltindilaurate - in healing electro-fluid plays an important role in the self-healing of soft materials. To avoid ambiguity, we have modified “self-healing blood” to “blood”. We have revised throughout the manuscript.

Soft robots are susceptible to cuts from sharp objects due to their construction from soft materials (S. Terryn, et al., Self-healing soft pneumatic robots. Sci. Robot. 2, eaan4268, 2017; S. Terryn, et al., A review on self-healing polymers for soft robotics. Mater. Today 47, 187-205,

2021). Previous papers have not proposed that soft robots take active measures to deal with damages on their own after being damaged. But the fact is that humans will take active measures to protect themselves when injured. Here, we propose the concept of self-protection in the expectation of realizing the robot's autonomous detection and taking active measures to heal the damage. Self-protection behavior is defined as a robot's ability to detect damage and then actively take protective measures to achieve rapid self-healing. We have added the information in Main Text (Marked in red on page 4).

In page 4, we have added

where the ability of a robot to detect damage and then actively take protective measures to achieve rapid self-healing is termed as self-protection behavior.

Blood plays an important role in the body's self-healing. If our hand is wounded, the blood clots in a few minutes, and after a few days, the clot peels off, new skin grows there, and the wound disappears. In other words, the wound is healed by the blood and the newly grown tissue. To avoid misunderstanding, we have removed these descriptions from the Abstract.

Comment 2

2, One feature that the authors highlighted is the dynamic performance of their soft robotic system. However, I cannot help noticing that the EHD pump takes quite a big volume fraction compared to the soft actuator. For a given hydraulic pump with a fixed pressure-flow diagram, if we reduce the soft actuator size, the required amount of liquid to inflate decreases, and the dynamic performance will improve. However, a very small soft actuator will also significantly limit the application scenarios for fluidic soft robots, especially considering the integration advantages for fluidic transmission.

In the paper, the authors did not exclude this important factor, but directly compared with other soft or stretchable pumps (in Extended data figure 1). Is this time constant an intrinsic feature of the pump? Or a feature of the actuator or combined? The authors need to consider the potential factors to provide a fair comparison.

I am not saying making small hydraulic actuators is not useful, but the expressions feel like it

is an intrinsic feature of the system independent of other system parameters. If the dynamic performance is so important for the authors, I highly recommend the authors to show the dynamic responses of their soft pump and the typical pressure-flow rate curve so other researchers in the soft robotic community can understand the potential of this pumping system. And if the authors insist on such modular soft robot systems with a big pump and small actuator, it is still ok. But then the authors are encouraged to explore their applications. For example, maybe demonstrate a mechanical display for blind people or a tactile interface between computer and human. I am sure there is potential and the authors can better promote their work to a bigger audience.

Answer 2

Thank you for your valuable comment. Soft EHD pumps can be customized to fit the scale of the actuator and the robot. Actually our pump and actuator are matched in size, it is not a big pump and small actuator, for example, in the bending robot (Fig. 1a), the volume of actuator is 24.2 cm^3 while the volume of pump is 20.2 cm^3 . But in the previous reported soft fluidic robots, the rigid body pumps would take up more volume and weight than ours (M. A. Bell, B. Gorissen, K. Bertoldi, J. C. Weaver, R. J. Wood, A Modular and Self-Contained Fluidic Engine for Soft Actuators, *Adv. Intell. Syst.* 4, 2100094, 2022; A. D. Marchese, C. D. Onal, D. Rus, Autonomous Soft Robotic Fish Capable of Escape Maneuvers Using Fluidic Elastomer Actuators, *Soft Robot.* 1, 75-87, 2014). Also the response speed of their system is much slower than the integration of our soft EHD pump (ours $< 0.25 \text{ s}$, others $\sim 4 \text{ s}$).

In fact, the EHD pump tightly integrated with the actuator could reduce the flow time, allowing for rapid deformation of the actuator. In case of a small piping connection of EHD pump and actuator, the response of the system would be much slower ($\sim 20 \text{ s}$, in reference (V. Cacucciolo, H. Nabaee, K. Suzumori, H. Shea, Electrically-Driven Soft Fluidic Actuators Combining Stretchable Pumps With Thin McKibben Muscles, *Front. Robot. AI* 6:146, 2020) the author said: A major limitation of these devices in their current form is the response time. We estimated a force rise time of 14 s and a fall time of 6.6 s .) We integrate the EHD pump with the actuator to achieve fast response of the system, which is a big advantage. For example,

the gripper (Fig. 4b) and the mechanical sieve (Fig. 4c) fully integrate the EHD pumps with the actuators, which can achieve fast actuation.

The response time of the system is a feature of the combined system and an important performance indicator of the robot. To provide a more fair and comprehensive comparison, we have added stretchability, force, actuation stroke, and power consumption. We have added the information in Supplementary information (Marked in red on Supplementary information page 21).

In Supplementary information page 17, we have revised

Supplementary Table 1 | Comparison of different EHD pumps when the pumps are embedded into robots or actuators.

	Actuation response	Stretchability	Force	Actuation stroke	Power consumption
Stretchable pump ¹⁶	30 s	Yes	\	~ 40°	~ 0.1 W
Electro-conjugate fluid pump ¹⁸	10 s	No	~ 0.18 N	~ 1.5 mm	~ 0.15 W
Electro-conjugate fluid pump ¹⁵	1.26 s	No	0.0066 N	~ 30°	\
Soft electronic pump ¹⁷	1 s	Yes	\	~ 8 mm	~ 3.6 W
Soft EHD pump (This study)	< 0.25 s	Yes	> 0.8 N	~ 14 mm ~ 85°	~ 6 W

According to your suggestions, we have added dynamic response, pressure, and flow rate curves of the soft EHD pump. We have added the information in Main Text and Supplementary information (Marked in red on page 6 and Supplementary information page 8).

In page 6, we have added

Supplementary Fig. 7 shows the pressure, flow rate, response time, and lifetime test of a soft

EHD pump with two electrode pairs.

In Supplementary information page 8, we have added

Supplementary Fig. 7 | Performance tests of a soft EHD pump with two electrode pairs.

a, Pressure-voltage curve. **b**, Flow rate-voltage curve. **c**, Response time curve. The response time (peak time) of the soft EHD pump is ~ 0.2 s.

According to your suggestions, we developed a small interface system that can be used for haptics and heat. We wore this system on a person's finger to give the finger a sense of touch and heat. Since human senses cannot be shown through video, we shown the sense of touch and heat through data testing. Combining a pump and an actuator is beneficial for this feature because haptics usually require fast actuation of the actuator. We have added the information in Main Text (Marked in red on pages 15 and 16).

In page 15, we have added

E-skin can be used for heat generation, while fluidic actuators can be used to generate tactile sensations. Combining the heating function of the E-skin and the actuation function of the fluidic actuator, we develop a tactile and thermal system (Fig. 4d). The system can give fingers tactile and thermal sensation (Fig. 4d), demonstrating the potential of soft fluidic robots for wearable devices.

In page 16, we have added

Fig. 4 | d, Tactile and thermal system.

Comment 3

3, The self-healing capability of the soft actuator and the filling liquid is interesting but the investigation is a bit sloppy. Although the concept was reported in the previous publication, the authors made improvements so it can adapt to “large area” damage. (How large is large here?) However, the damage shown in Fig.3d on the real robotic actuator is quite different from punching a hole, am I right? There are so many ways the soft actuators can get damaged and the demonstration is too preliminary. After reading the experiments, I still don't fully understand or don't have enough experimental evidence to evaluate to what extent the self-healing will work and when it will not? Does self-healing impact on the dynamic performances of the soft robotic systems? The results are thin compared to the authors' strong claims. Personally, I cannot reach solid conclusions but only find the method promising. Maybe the authors can consider working on this.

Answer 3

Thank you for your helpful comment. Small-area damage refers to material damage where the fracture surface of damage is in contact (Figure 2-1A), and large-area damage refers to material damage where the fracture surface of damage is not in contact (Figure 2-1B, C).

Figure 2-1

Previous studies on self-healing can only achieve self-healing of small areas, and large-areas self-healing is a challenge. It is difficult to fill in damage where contact surfaces are not connected because elastomer materials cannot grow themselves.

Undoubtedly, our proposed large-areas-damage self-healing has some limitations. Healing electro-fluid exposed to air can be cured to fill the damage, but if the damage is very large, it is difficult to heal the elastomer with a greater area of curing electro-fluid. After many tests, we find that self-healing can be achieved at a distance of less than 5 cm of the fracture surface in damage. When the distance between the broken fracture surfaces is greater than 5 cm, the electro-fluid curing tends to form holes that could lead to failure. We have added the information in Main Text (Marked in red on page 11).

In page 11, we have added

Undoubtedly, our proposed large-areas-damage self-healing has some limitations. Healing electro-fluid exposed to air can be cured to fill the damage, but if the damage is very large, it

is difficult to heal the elastomer with a greater area of curing electro-fluid. After many tests, we find that self-healing can be achieved at a distance of less than 5 cm of the fracture surface in damage. When the distance between the broken fracture surfaces is greater than 5 cm, the electro-fluid curing tends to form holes that could lead to failure.

The damage on the real robotic actuator similar to that in Figure 2-2, which is also a kind of large-area damage where its fracture surface is not in contact. Punching a hole is a kind of large-area damage (Figure 2-1C).

Damage with no contact on
the fracture surface

Figure 2-2

To compare the mechanical properties of pristine, damaged, and self-healed soft elastomer materials, we conducted the tensile stress-strain test. Damaged samples (hole 3-4 mm) were easily pulled off and had degraded mechanical properties, and after self-healing they could achieve similar mechanical properties to the pristine samples (Supplementary Fig. 16b). We have added the information in Main Text and Supplementary information (Marked in red on page 11 and Supplementary information page 17).

In page 11, we have added

To further evaluate their mechanical properties, we tested stress-strain curves of pristine, 250 °C (15 s), damaged (hole 3-4 mm), and self-healed samples using Silicone 5A as an example.

Damaged samples were easily pulled off and had degraded mechanical properties, and after self-healing they could achieve similar mechanical properties to the pristine samples (Supplementary Fig. 16b).

In Supplementary information page 17, we have added

Supplementary Fig. 16 | Tensile tests. To test the mechanical properties of samples (silicone 5A), uniaxial tensile tests were conducted at an ASTM D638 (Type IV) universal test machine with a crosshead speed of 50 mm/min. **b**, Tensile stress-strain curves of pristine, damaged, and self-healed samples.

Comment 4

4, The demonstrations in Fig.4 are nice but I would not consider it as a contribution in this work. They have been reported in other publications, and are quite far from real-world applications. But they certainly show the systems function well and I appreciated that.

Answer 4

Thank you so much for your appreciation, these demos show the advantages of this integration we propose. Based on your suggestion, we developed a small interface system that can be used for haptics and heat. We wore this system on a person's finger to give the finger a sense of touch and heat. Since human senses cannot be shown through video, we shown the sense of touch and heat through data testing. Combining a pump and an actuator is beneficial for this feature because haptics usually require fast actuation of the actuator. We have added the information in Main Text (Marked in red on pages 15 and 16).

In page 15, we have added

E-skin can be used for heat generation, while fluidic actuators can be used to generate tactile sensations. Combining the heating function of the E-skin and the actuation function of the fluidic actuator, we develop a tactile and thermal system (Fig. 4d). The system can give fingers tactile and thermal sensation (Fig. 4d), demonstrating the potential of soft fluidic robots for wearable devices.

In page 16, we have added

Fig. 4 | d, Tactile and thermal system.

Comment 5

5, In fact, there is quite some nice engineering work done. For example, the authors develop multi-material 3d printing to improve the electrode design. I am wondering how they are compared to the metal electrodes in [17]. Is there any problem with the long time performance of the conductive TPU? And to what extent is the pump can be deformed and still functional? Does soft material generate mismatched assembly positions between the cone electrodes and the pore positions, especially under the dynamic flow?

Answer 5

Thank you so much for your appreciation. In fact, our previous work [17] (Wei Tang, et al. Customizing a self-healing soft pump for robot. *Nature Communications*, 12:2247, 2021) did not use metal electrodes, but rather used silicone rubber as electrodes (Conductive silicone 50A and Conductive silicone 60A). The electrodes in [17] were fabricated using silicone rubber casting methods, which is difficult to achieve large-scale integrated manufacturing. As a result,

it is difficult to further improve system response speed of EHD pumps in [17]. In order to better fabricate more electrode pairs, in this work, we propose to use multi-material 3D printing to fabricate electrodes. This work outperformed our previous work [17], with a fourfold increase in system response speed.

We test the lifetime of the soft EHD pumps at an applied voltage of 12 kV for 6 h (Supplementary Fig. 7d), illustrating the durability and reliability of the soft EHD pumps. It is worth mentioning that the electrodes would be passivated and then stabilized when a high voltage was supplied for a longer period of time, causing the pressure to drop and then stable [17]. We have added the information in Main Text and Supplementary information (Marked in red on page 5 and Supplementary information page 8).

In page 5, we have added

Supplementary Fig. 7 shows the pressure, flow rate, response time, and lifetime test of a soft EHD pump with two electrode pairs.

In Supplementary information page 8, we have added

Supplementary Fig. 7 | Performance tests of a soft EHD pump with two electrode pairs.

d, Lifetime test of the soft EHD pump. We test the lifetime of the soft EHD pumps at an applied voltage of 12 kV for 6 h, illustrating the durability and reliability of the soft EHD pumps. It is

worth mentioning that the electrodes would be passivated and then stabilized when a high voltage was supplied for a longer period of time, causing the pressure to drop and then stable¹⁷.

As long as the electrode pair is not shorted, soft EHD pumps can be deformed and still function, but if the deformation causes dielectric breakdown in the pumps, they will fail. Our experimental tests show that the dynamic flow of the electro-fluid does not have a significant effect on the electrode pairs. We have added the information in Main Text (Marked in red on page 19).

In page 19, we have added

As long as the electrode pair is not shorted, soft EHD pumps can be deformed and still function, but if the deformation causes dielectric breakdown in the pumps, they will fail. Our experimental tests show that the dynamic flow of the electro-fluid does not have a significant effect on the electrode pairs.

Comment 6

Overall I think the paper has some solid engineering but with an unfit story. It has many claims that are not well-defined and investigated in detail, and I am not sure some of the nice features are compatible. The authors are not well explained what are the compromises for those performances and what are the associated limitations. Maybe it is more acceptable in the material community but based on the current version, I would recommend it for a more specialized journal (or a material science journal).

Answer 6

Thank you for your comment. We revised the story of the manuscript. According to your suggestions, we defined and investigated some claims in details.

Our work addresses several key challenges for fluid-driven soft robots.

(1) Fluidic power sources

In paper (S. I. Rich, Robert J. Wood, Carmel Majidi. Untethered soft robotics. *Nature*

Electronics, 1:102-112, 2018): “One limitation of pneumatic and hydraulic actuators is the requirement of an external pump to pressurize and depressurize the working fluid”, “Several groups have proposed innovative methods to circumvent the need for this equipment, effectively removing the pneumatic tether from these actuators”, “These techniques effectively remove the pneumatic tether but introduce new challenges like slow response speed”.

There are many other papers that point out that embedding fluidic power sources in soft robots or actuators is a persistent challenge. Our previous work [17] (Wei Tang, et al. Customizing a self-healing soft pump for robot. *Nature Communications*, 12:2247, 2021) proposed the prototype of the soft EHD pump, but it did not work well when embedded in the robot or actuator and the system response speed was slow.

In this work, we propose a novel electrode pair dubbed a conical-array-porous-plate electrode pair (Fig. 1e) to obtain high-performance outputs of the soft EHD pump. Since the conical array is difficult to fabricate by conventional casting method, we propose using multi-material 3D printing to construct the soft EHD pump, with conductive TPU for the electrodes and non-conductive TPU for the slots and supports (Fig. 1f and Supplementary Fig. 5). The high-performance soft EHD pumps enable high-speed actuation and large deformation of untethered soft fluidic robots, which addresses the challenge of fluidic power sources in soft robots.

(2) Large-area self-healing

In paper (J. Kang, J. B.-H. Tok, Zhenan Bao. Self-healing soft electronics. *Nature Electronics*, 2:144-150, 2019): “**Large-area.** When the size of damages is larger than the thickness of layers, it is challenging to realize an efficient self-healing process. Therefore, the self-healing process is currently limited to only small-area damages.”

There are many other papers that point out large-area self-healing is a persistent challenge. In this work, we propose a new kind of healing electro-fluids (Fig. 3a and Methods). When exposed to air, the healing electro-fluid can cure into a healed film, which possesses excellent

stretchability (>1200%, Fig. 3b) and ultra-strong adhesion properties (Fig. 3c). We tested the repair properties of the healing electro-fluid on different silicone rubbers which are widely used in soft robots and found that the fluid could fill the large damage with no contact on the fracture surface, i.e., large-area damage, and achieve self-healing (Fig. 3d), which addresses the challenge of large-area self-healing in soft robots.

(3) Self-protection

Previous research has not realized the ability of soft robots to detect damage, self-judgment, and actively take protective measures to achieve rapid self-healing when damaged. In this work, we develop multi-functional E-skins to endow robots intelligence, making soft robots realize a series of self-protection behaviors, including self-sensing, self-judgment, and self-healing for rapid self-healing.

The development of soft robotics is a topic that *Nature Communications* is interested in, and *Nature Communications* has recently launched a Collection (<https://www.nature.com/collections/bgfdjghifc>, Soft Robotics: Sensing, Actuation, and Integration, see Figure 2-3 in next page) on soft robotics. Our work falls within the scope of *Nature Communications*, and more specifically, within the scope of this Collection (Soft Robotics: Sensing, Actuation, and Integration).

Collection

Soft Robotics: Sensing, Actuation, and Integration

Submission status

Open

Submission deadline

30 September 2023

Inspired by invertebrate organisms like jellyfish and earthworm, the research of soft robotics as alternatives to traditional rigid robots has made tremendous progress in recent years. Scientists are looking for solutions not only mimicking highly versatile locomotion of marine animals but also embedding multiple sensing and actuating functionalities into soft materials that enable adaptability to the environment. More importantly, soft robotic technologies hold great promises for biomedical applications including wearable, prosthetic robots, and miniaturized surgical devices.

In this collection at *Nature Communications*, we aim to bring together cutting-edge soft robotics research crossing multidisciplinary areas. Topics of interest include but are not limited to the following:

- Tactile/flexible sensors for soft robotic applications
- Fluid-driven/magnetic-driven soft robot
- Pneumatic muscles
- Soft gripper/walker/jumper/swimmer
- Origami robot
- Biomedical applications of soft robot

Figure 2-3

Reviewer #3 (Remarks to the Author):

In summary: The manuscript reported a bio-inspired design strategy for self-protection soft fluidic robots and the soft robots integrate EHD pumps, actuators, self-healing electro-fluids, and E-skins. Poor fluidic power source, large area self-healing, and active self-protection of soft robots are persistent challenges in soft robotics. The important contributions made by the authors to address these challenges are exciting. The authors' development of rapid large area self-healing and the proposed concept of self-protection can greatly advance the development of soft robotics. Additionally, the authors significantly increased the EHD pump's response speed, which is far-reaching for the solution of actuating fluidic soft robotics. The manuscript is suitable for publication in Nature Communications after the following questions are addressed.

Response:

Thank you very much for taking your valuable time to review our manuscript in your busy schedule. We feel great thanks for your professional and valuable comments to improve our manuscript. According to your constructive comments, we have made corresponding modifications and added new experimental data/results. Point-by-point corrections/responses are listed below. Thank you for your positive comments!

Comment 1

1. The hardness and resistivity of the conductive TPU used for 3D printing EHD pumps should to be characterized.

Answer 1

Thank you for your helpful comment. The hardness of conductive TPU is 90 A, and the resistances of the conical array electrode and porous plate electrode printed by conductive TPU were $\sim 4\text{--}10\text{ k}\Omega$ and $\sim 0.3\text{--}0.9\text{ k}\Omega$, respectively. We have added the information in Main Text (Marked in red on pages 18-19).

In pages 18-19, we have added

Conductive TPU (Eel, 90A, NinjaTek) and non-conductive TPU (60A, NinjaTek) were used as the printing materials for the pump.

The 3D printing process was shown in Supplementary Fig. 5a, and the photos of the printed parts were shown in Supplementary Figs. 5b-g, where the resistances of the conical array electrode and porous plate electrode were $\sim 4\text{--}10\text{ k}\Omega$ and $\sim 0.3\text{--}0.9\text{ k}\Omega$, respectively.

Comment 2

2. Please explain why the conical electrode the authors propose is better than the cylindrical electrode the authors previously reported.

Answer 2

Thank you for your kind comment. Compared to the cylindrical electrode, the conical electrode can produce a stronger electric field strength. We have added the information in Main Text (Marked in red on page 5).

In page 5, we have added

The EHD flow effect is enhanced by using the conical array as the positive electrode of the pump and a porous plate as the negative electrode because the conical array can produce a stronger electric field than the previously described cylindrical electrode¹⁷.

Comment 3

3. The authors should add experiments and data on the system response of the EHD pumps.

Answer 3

Thank you for your valuable comment. We have added experiments and data on the system response of the EHD pumps, as shown in Supplementary Fig. 6. We have added the information in Main Text and Supplementary information (Marked in red on page 5 and Supplementary information page 7).

In page 5, we have added

Our experimental results indicate that the soft EHD pump created by the conical-array-porous-plate electrode pairs and multi-material 3D printing method achieves a striking system actuation speed (when the pump is embedded into a robot, < 0.25 s, Supplementary Fig. 6)

In Supplementary information page 7, we have added

Supplementary Fig. 6 | System actuation speed when a soft EHD pump is embedded into a robot. The robot is a contracting fluidic robot with a 20-g load. It takes ~ 0.25 s for the robot to reach the maximum stroke of 14 mm. Response time for other deformations is less than 0.25 s.

Comment 4

4. The authors should do a comprehensive response speed comparison of more EHD pumps.

Answer 4

Thank you for your valuable comment. We have extended SI Table 1, added stretchability, force, actuation stroke, and power consumption, and added more EHD pumps in the literature. We have added the information in Supplementary information (Marked in red on Supplementary information page 17).

In Supplementary information page 17, we have revised

Supplementary Table 1 | Comparison of different EHD pumps when the pumps are embedded into robots or actuators.

	Actuation response	Stretchability	Force	Actuation stroke	Power consumption
Stretchable pump ¹⁶	30 s	Yes	\	~ 40°	~ 0.1 W
Electro-conjugate fluid pump ¹⁸	10 s	No	~ 0.18 N	~ 1.5 mm	~ 0.15 W
Electro-conjugate fluid pump ¹⁵	1.26 s	No	0.0066 N	~ 30°	\
Soft electronic pump ¹⁷	1 s	Yes	\	~ 8 mm	~ 3.6 W
Soft EHD pump (This study)	< 0.25 s	Yes	> 0.8 N	~ 14 mm ~ 85°	~ 6 W

Comment 5

5. In Figure 2, why is there a negative angle in the twisting angle of the twisting actuator, and how is it defined?

Answer 5

Thank you for your helpful comment. We have defined the twisting angle. We have added the information in Main Text (Marked in red on page 9).

In page 9, we have added

The angle between the two battens is zero while the robot is not in motion; it is recorded as a positive twisting angle when it generates counterclockwise twisting motion, and as a negative twisting angle when it generates clockwise twisting motion.

Comment 6

6. Please clarify the advantages and disadvantages of the author's two methods for creating E-skin.

Answer 6

Thank you for your kind comment. E-skin without a seal can be refilled with liquid metal when there is a problem with the E-skin, but the exposed liquid metal is easily disturbed. E-skin with a seal is convenient to use, however it is challenging to find E-skin issues when they occur. We have added the information in Main Text (Marked in red on page 19).

In page 19, we have added

E-skin without a seal can be refilled with liquid metal when there is a problem with the E-skin, but the exposed liquid metal is easily disturbed. E-skin with a seal is convenient to use, however it is challenging to find E-skin issues when they occur.

Comment 7

7. How is the self-healing liquid's curing time measured, and how is the curing curve created?

Answer 7

Thank you for your valuable comment. A temperature box with adjustable temperature was used to test the healing electro-fluid's curing time at different temperature. Several healing electro-fluids were positioned on the silicone sheets, and the silicone sheets were put in the temperature box. The self-healing region was periodically probed with a rod to evaluate whether the fluids had cured, and a timer was utilized to record the self-healing time at the same time. We have added the information in Main Text (Marked in red on page 19).

In page 19, we have added

A temperature box with adjustable temperature was used to test the healing electro-fluid's curing time at different temperature. Several healing electro-fluids were positioned on the silicone sheets, and the silicone sheets were put in the temperature box. The self-healing region was periodically probed with a rod to evaluate whether the fluids had cured, and a timer was utilized to record the self-healing time at the same time.

Comment 8

8. The authors seem to have different pumps for each soft robot, please clarify in detail the pumps used for different robots.

Answer 8

Thank you for your valuable comment. The pump depends the electrodes. Two different electrodes (Supplementary Figs. 5b, c, and Supplementary Figs. 5d, e) were used in this paper. We clarify the electrodes for different robots (Marked in red on pages 9 and 13).

In page 9, we have added

a, Bending motion of the fluidic robot. The robot's electrodes are shown in Supplementary Figs. 5b and c.

d, Twisting motion of the fluidic robot. The robot's electrodes are shown in Supplementary Figs. 5d and e.

g, Contracting motion of the fluidic robot. The robot's electrodes are shown in Supplementary Figs. 5d and e.

In page 13, we have added

a, Architecture of a contracting soft fluidic robots. The robot's electrodes are shown in Supplementary Figs. 5d and e.

Comment 9

9. The sizes of the two types of beads used in the screening experiments need to be clarified. Also, please describe in detail how the mechanical sieve operates.

Answer 9

Thank you for your helpful comment. The diameters of large beads and small beads are 10 mm and 5 mm, respectively (Marked in red on page 27). We have added more information of operation of mechanical sieve (Marked in red on page 28).

In page 27, we have added

The mechanical sieve (Fig. 4c) consisted of a fluidic robot, a screening container with a size-selective gate, large beads (diameter 10 mm), small beads (diameter 5 mm), and a collection container.

In page 28, we have added

To operate the mechanical sieve, we built a control circuit using an Arduino, three high-voltage power converters, and a battery, where the output intervals and frequency are controlled by a program. The mechanical sieve screening process is shown in Fig. 4c, noting the actuator in the activated state as 1, at which point the actuator elongates, and the unactivated state as 0, with a actuation interval of 0.5 s. Four intermediate consecutive moments were selected for analysis (Fig. 4c): at 3.5 s, the states of the three actuators were 0, 1, and 1, respectively, and the screening container was tilted toward actuator 1; at 4 s, actuator 1 was activated, actuator 2 was unactivated, actuator 3 remained in place, and the screening container was tilted toward actuator 2, which put small beads into the collection container through the size-selective gate; at 4.5 s, actuator 2 was activated, actuator 3 was unactivated, actuator 1 remained in place, and the screening container was tilted towards actuator 3 to mix the beads; at 5 s, it returned to the state of 3.5 s, mixing beads and starting the next cycle. Bead screening can be finished in ~ 28 s, and manual control was then used to complete this process.

REVIEWER COMMENTS

Reviewer #1 (Remarks to the Author):

I appreciate the careful revision of the manuscript. All my remarks and questions are addressed in full. I recommend the manuscript for publication.

Reviewer #2 (Remarks to the Author):

I appreciate the authors' detailed response and careful explanation, which help me to better understand the paper. I am also very grateful for the authors' willingness to do extra work to improve the quality of the paper, which helps me gain confidence that this work will eventually be published.

Comment 1:

I am glad that the authors removed some controversial statements. Although I do not have much of a medical background, their description of the wound healing process is now acceptable to me.]

Comment 2:

I appreciate the authors' comments, especially a new table comparing different soft pumps. I really like the authors' explanation of their compact system integration, which contributes significantly to the improved dynamic performance, and I highly recommend adding it to the main text.

I still have a few questions. Since the dynamic performance is a crucial contribution of this work (the author also emphasizes it in the abstract), it is very important to explain the mechanism for such an improvement. In the supplementary table, the authors compare different EHD micropumps, removing the influence of the actuator size and the connection pipes. However, in this table there is quite a difference in response time. And I would like the authors to provide their argument for the differences. For example, all the pumps listed have a similar mechanism of injecting electric charges directly into the dielectric fluid and accelerating the fluid by the electric field and viscous drag. Does the difference in dynamic response have anything to do with this process? My second guess would be the flexibility of

the pump itself. The choice of materials can increase the internal "fluidic" capacitance that attenuates the signal. Are there other factors that could explain the difference? I suggest the authors also include the drive voltage in the comparison table, as it seems to be related to the response time.

Comment 3:

Thanks for the detailed explanation of "large-area" damage and the comments on the limitation. The added materials really helped me understand the capabilities of this technology. I am very impressed with the self-healing ability. But there are two concerns: 1, I suspect that methyltracetoxy silane and dibutyltin dilaurate are more chemically similar to silicone rubber, so they will accumulate on the silicone rubber over time. How does the silicone swell over time in this electrofluid? It would be helpful to know the swelling ratio and time associated with this fluid mixture.

2, this also raises concerns about the long-term stability of these healing capabilities. Is the soft robot still able to heal itself a week after being filled with the self-healing fluid?

Comments 4 and 5

I appreciate the new demonstration and experiments to evaluate the long-term stability.

Comment 6

Thank you for explaining the contribution of this paper. I find them very convincing. I strongly suggest the authors to add a sentence to explain the "large-area self-healing" (the size of damage is larger than the thickness of layers) and "self-protection" (actively take protective measures to achieve rapid self-healing when damaged) when it first appeared in the main text. I think this would be helpful to readers, especially considering the wide readership of Nature Communications.

One additional comment:

The authors did many demonstrations, and not all of them with fully integrated systems. (e.g. some with piles and some with electrical wires) Can the authors provide a list of their demonstrations about their level of system integration? It is not very easy to see from the figures because the packaging is done naively.

Reviewer #3 (Remarks to the Author):

The authors have addressed the comments I raised in the previous round of review. I now recommend it for publication.

Detailed Responses to Reviewers

Reviewer #1 (Remarks to the Author):

I appreciate the careful revision of the manuscript. All my remarks and questions are addressed in full. I recommend the manuscript for publication.

Response:

Thank you very much for taking your valuable time to review our manuscript in your busy schedule. It is of great honor for us to get your helpful comments to improve our manuscript.

We would like to express our heartfelt gratitude for your effort to improve our work. We also thank you for your recognition of our work.

Thanks again!

Reviewer #2 (Remarks to the Author):

In summary:

I appreciate the authors' detailed response and careful explanation, which help me to better understand the paper. I am also very grateful for the authors' willingness to do extra work to improve the quality of the paper, which helps me gain confidence that this work will eventually be published.

Response:

Thank you very much for taking your valuable time to review our manuscript in your busy schedule. We feel great thanks for your professional and valuable comments to improve our manuscript. According to your constructive comments, we have made corresponding modifications and added new experimental data/results. Point-by-point corrections/responses are listed below. Thank you for your positive comments!

Comment 1

I am glad that the authors removed some controversial statements. Although I do not have much of a medical background, their description of the wound healing process is now acceptable to me.]

Answer 1

Thank you for recognizing our revised manuscript!

Comment 2

I appreciate the authors' comments, especially a new table comparing different soft pumps. I really like the authors' explanation of their compact system integration, which contributes significantly to the improved dynamic performance, and I highly recommend adding it to the main text.

I still have a few questions. Since the dynamic performance is a crucial contribution of this work (the author also emphasizes it in the abstract), it is very important to explain the

mechanism for such an improvement. In the supplementary table, the authors compare different EHD micropumps, removing the influence of the actuator size and the connection pipes. However, in this table there is quite a difference in response time. And I would like the authors to provide their argument for the differences. For example, all the pumps listed have a similar mechanism of injecting electric charges directly into the dielectric fluid and accelerating the fluid by the electric field and viscous drag. Does the difference in dynamic response have anything to do with this process? My second guess would be the flexibility of the pump itself. The choice of materials can increase the internal "fluidic" capacitance that attenuates the signal. Are there other factors that could explain the difference? I suggest the authors also include the drive voltage in the comparison table, as it seems to be related to the response time.

Answer 2

Thank you for your positive comment. We have added the table to the main text (Marked in red on page 8, Table 1).

In our experiments, we found that the dynamic response was related to the flow speed and flow rate of the liquid, did not seem to be related to the process of charge injection or to the flexibility of the pump itself.

In Table 1, there are five different types of pumps: (i) stretchable pump¹⁶; (ii) electro-conjugate fluid pump¹⁸; (iii) electro-conjugate fluid pump¹⁵; (iv) soft electronic pump¹⁷; and (v) soft EHD pump (this study), where the electrodes of (i) are planar electrode structures and electrodes of (ii)-(v) are needle-hole electrode structures. Since the needle-hole electrode structure has a stronger ability to actuate the fluid flow compared to the planar electrode structure, (ii)-(v) have a larger flow rate than (i), and therefore the system responses of (ii)-(v) are faster than (i). (iii) integrated the actuator more tightly with the liquid reservoir, so (iii) has a faster system response than (ii). Since (iv) integrated more electrode pairs, the relative flow rate is increased, resulting in (iv) possessing a faster system response than (iii). The reason that (v) is better than (iv) is that, on the one hand, the conical electrode structure we propose can produce a stronger electric field strength than the cylindrical electrode structure, which increases the flow speed, and, on the other hand, we use 3D printing method to fabricate the electrodes so that more electrode pairs can be integrated, which increases the flow rate. We

have added the information in main text (Marked in red on page 21).

Different scales of pumps have different actuation voltages, the larger the pump size, the larger the actuation voltage tends to be, and there is no definite relationship with response time. We have added the actuation voltages to the Table (Marked in red on page 8, Table 1).

In page 8, we have added

Table 1 | Comparison of different EHD pumps when the pumps are embedded into robots or actuators.

	Actuation response	Stretchability	Force	Actuation stroke	Power consumption (actuation voltage)
Stretchable pump ¹⁶	30 s	Yes	\	~ 40 °	~ 0.1 W (8 kV)
Electro-conjugate fluid pump ¹⁸	10 s	No	~ 0.18 N	~ 1.5 mm	~ 0.15 W (6 kV)
Electro-conjugate fluid pump ¹⁵	1.26 s	No	0.0066 N	~ 30 °	\ (4.5 kV)
Soft electronic pump ¹⁷	1 s	Yes	\	~ 8 mm	~ 3.6 W (16 kV)
Soft EHD pump (This study)	< 0.25 s	Yes	> 0.8 N	~ 14 mm ~ 85 °	~ 6 W (14 kV)

In page 21, we have added

In Table 1, there are five different types of pumps: (i) stretchable pump¹⁶; (ii) electro-conjugate fluid pump¹⁸; (iii) electro-conjugate fluid pump¹⁵; (iv) soft electronic pump¹⁷; and (v) soft EHD pump (this study), where the electrodes of (i) are planar electrode structures and electrodes of (ii)-(v) are needle-hole electrode structures. Since the needle-hole electrode structure has a

stronger ability to actuate the fluid flow compared to the planar electrode structure, (ii)-(v) have a larger flow rate than (i), and therefore the system responses of (ii)-(v) are faster than (i). (iii) integrated the actuator more tightly with the liquid reservoir, so (iii) has a faster system response than (ii). Since (iv) integrated more electrode pairs, the relative flow rate is increased, resulting in (iv) possessing a faster system response than (iii). The reason that (v) is better than (iv) is that, on the one hand, the conical electrode structure we propose can produce a stronger electric field strength than the cylindrical electrode structure, which increases the flow speed, and, on the other hand, we use 3D printing method to fabricate the electrodes so that more electrode pairs can be integrated, which increases the flow rate.

Comment 3

Thanks for the detailed explanation of "large-area" damage and the comments on the limitation. The added materials really helped me understand the capabilities of this technology. I am very impressed with the self-healing ability. But there are two concerns:

1, I suspect that methyltracetoxy silane and dibutyltin dilaurate are more chemically similar to silicone rubber, so they will accumulate on the silicone rubber over time. How does the silicone swell over time in this electrofluid? It would be helpful to know the swelling ratio and time associated with this fluid mixture.

2, this also raises concerns about the long-term stability of these healing capabilities. Is the soft robot still able to heal itself a week after being filled with the self-healing fluid?

Answer 3

Thank you for your valuable comment. In our previous experiments, we have found the process of electrofluid swelling of silicone rubber, and the relevant reference also mentions the swelling problem (Nature 572, 516-519, 2019). This is due to that silicone rubber is a kind of porous material and some electrofluid can seep into the silicone pores, thus swelling the silicone rubber. In our previous tests, we found that silicone rubber tended to stabilize when swelling reached 5%, which had little effect on experimental testing. We previously tested the swelling ratios of several electrofluids commonly used in our experiments with silicone rubber. Swelling times to ~ 5% for different electrofluids are: linalyl acetate functional liquid, ~ 2 h; dibutyl sebacate

functional liquid, ~ 20 days; and Fluorinert FC-40 functional liquid, > 50 days. Improvement of electrofluid performance is an important direction for our follow-up research. We have added the information in main text (Marked in red on page 23).

The healing electrofluid would fail over time in a closed silicone rubber chamber, due to the fact that the healing electrofluid we developed is not an industrial grade product. The time to failure is related to the temperature and environment in which it is kept. Our previous experiments shown that the time to failure of the healing electrofluid was about 10 days at room temperature in the laboratory. For soft robots, it is still able to heal itself a week after being filled with the healing electrofluid. However, if the robot is to be actuated and is placed horizontally, the swelling process would produce air bubbles; these air bubbles are present inside the EHD pump and would cause the robot's actuation to fail. At this point, the air bubbles must be pumped out in order to achieve actuation. If the robot is placed vertically, the air bubbles collect at the actuator and there are no air bubbles inside the EHD pump, allowing the robot to be actuated. We have added the information in main text (Marked in red on page 23).

In page 23, we have added

The time to failure of the healing electrofluid was about 10 days at room temperature in the laboratory. In addition, the electrofluid would swell silicone rubber. Silicone rubber tended to stabilize when swelling reached 5%, which had little effect on experimental testing. Swelling times to ~ 5% for different electrofluids are: linalyl acetate functional liquid, ~ 2 h; dibutyl sebacate functional liquid, ~ 20 days; and Fluorinert FC-40 functional liquid, > 50 days. Improvement of electrofluid performance is an important direction for our follow-up research.

Comment 4-5

I appreciate the new demonstration and experiments to evaluate the long-term stability.

Answer 4-5

Thank you for your positive comment.

Comment 6

Thank you for explaining the contribution of this paper. I find them very convincing. I strongly suggest the authors to add a sentence to explain the "large-area self-healing" (the size of damage is larger than the thickness of layers) and "self-protection" (actively take protective measures to achieve rapid self-healing when damaged) when it first appeared in the main text. I think this would be helpful to readers, especially considering the wide readership of Nature Communications.

Answer 6

Thank you for your recognition of our work. We have added the information in main text (Marked in red on pages 3-5).

In page 3, we have added

In addition, it is worthwhile to conduct research on self-protection (robot actively takes protective measures to achieve rapid self-healing when damaged)

In page 4, we have revised

we design a class of self-protection soft fluidic robots with untethered actuation and self-protection behaviors (Fig. 1a), where the ability of a robot to detect damage and then actively take protective measures to achieve rapid self-healing is termed as self-protection behavior.

In page 5, we have added (large-area)

allowing for large-area self-healing (the size of damage is larger than the thickness of layers or the fracture surface of damage is not in contact) of soft materials (Fig. 1b).

In page 3, we have revised (small-area)

which requires a long self-healing time and only realizes small-area damages²⁰, that is, the size of damages is smaller than the thickness of layers or the damage fracture surface is in contact.

Comment 7

One additional comment:

The authors did many demonstrations, and not all of them with fully integrated systems. (e.g. some with piles and some with electrical wires) Can the authors provide a list of their demonstrations about their level of system integration? It is not very easy to see from the figures because the packaging is done naively.

Answer 7

Thank you for your helpful comment. We have added a table to show system integration level of demonstrations (Marked in red on Supplementary information page 23, Supplementary Table 3, and page 24).

In Supplementary information page 23, we have added

Supplementary Table 3 | System integration level of demonstrations.

	System integration level
Supplementary Video 1	Full integrated system (untethered)
Supplementary Video 2	Full integrated system (untethered)
Supplementary Video 3	Full integrated system (untethered)
Supplementary Video 4	Tethered
Supplementary Video 5	Tethered
Supplementary Video 6	Full integrated system (untethered)
Supplementary Video 7	Tethered
Supplementary Video 8	Full integrated system (untethered)
Supplementary Video 9	Full integrated system (untethered)

In main text page 24, we have added

where their system integration level can be seen in Supplementary Table 3.

Reviewer #3 (Remarks to the Author):

The authors have addressed the comments I raised in the previous round of review. I now recommend it for publication.

Response:

Thank you very much for taking your valuable time to review our manuscript in your busy schedule. It is of great honor for us to get your helpful comments to improve our manuscript.

We would like to express our heartfelt gratitude for your effort to improve our work. We also thank you for your recognition of our work.

Thanks again!

REVIEWERS' COMMENTS

Reviewer #2 (Remarks to the Author):

I really appreciate the authors' careful consideration and responses to my comments, and now I can recommend it for publication

Detailed Responses to Reviewers

Reviewer #2 (Remarks to the Author):

I really appreciate the authors' careful consideration and responses to my comments, and now I can recommend it for publication.

Response:

Thank you very much for taking your valuable time to review our manuscript in your busy schedule. It is of great honor for us to get your helpful comments to improve our manuscript.

We would like to express our heartfelt gratitude for your effort to improve our work. We also thank you for your recognition of our work.

Thanks again!